# Influence of Long-Term Wind Variability on the Storm Activity in the Caspian Sea

**Elizaveta Kruglova** [1,2,*] and **Stanislav Myslenkov** [1,2,3]

1   Department of Oceanology, Faculty of Geography, Lomonosov Moscow State University, GSP-1,
    Leninskie Gory, 119991 Moscow, Russia; stasocean@gmail.com
2   Shirshov Institute of Oceanology, Russian Academy of Sciences, Nakhimovsky pr. 36, 117997 Moscow, Russia
3   Hydrometeorological Research Centre of Russian Federation, Bolshoy Predtechensky Lane,
    123376 Moscow, Russia
*   Correspondence: lissavetaandin@gmail.com

**Abstract:** Wind and wave conditions are limiting factors for economic activity, and it is very important to study the long-term variability of storm activity. The main motivation of this research is to assess the impact of wind variability on the storm activity in the Caspian Sea over the past 42 years. The paper presents the analysis of a number of storms based on the results of wave model WAVEWATCH III and the Peak Over Threshold method. The mean, maximum, and 95th percentile significant wave heights were analyzed by season. The highest waves were in the Middle Caspian Sea in winter. Detailed interannual and seasonal analyses of the number and duration of storm waves were performed for the whole Caspian Sea and its separate regions. Positive significant trends were found in the whole sea. Significant positive trends in the number and duration of storms were found for the North and Middle Caspian. In the South Caspian, the trends were negative and not significant. High correlations were found between the number of storms and events with wind speed > 10–14 m/s and 95th percentile wind speed. Positive trends in the number of storms in the Middle Caspian were caused by positive trends in extreme wind situations.

**Keywords:** wind waves; wave modeling; number of storms; trend; WAVEWATCH III; Caspian Sea; wind variability





## 1. Introduction

The Caspian Sea is a very perspective and developing region. Intensive shipping, fishery, and oil and gas mining are carried out in this region [1]. Cargo turnover at seaports in the Caspian Sea in the first 8 months of 2022 was 4.2 million tons, including 2.1 million tons of dry cargo (+12.3% compared with the same period of the previous year) and 2.1 million tons of liquid cargo [2]. The PJSC LUKOIL Oil Company plans to start gas extraction at the field named after Kuvykin in the Caspian Sea in 2027 [3]. On the other hand, the hydrologic processes in the Caspian Sea strongly influence global climate change [4]. The assessment of climate change in storm activity and long-term wind variability is important for a wide variety of civil engineering and science tasks, ship and marine infrastructure safety, weather forecast, etc. [5].

The Caspian Sea is located on the border between Eastern Europe and Asia. This enclosed body of water extends strongly meridionally [6]. The Caspian Sea is a fully enclosed inland water body with a maximum depth of 1025 m. The northern part of the Caspian Sea is shallow (no more than 20 m deep). In this part of the Caspian Sea, development of the wave field is limited by depth. In the deeper Middle and South parts of the Caspian Sea, a significant role in wave development is the meridional stretching of the basin. Therefore, wave development at the western and eastern wind directions is limited by a fetch [5]. Given the varied topography of the seabed, as well as differences in physical and geographical conditions, it is customary to divide the Caspian Sea into

the North, Middle, and South Caspian Seas [7–9]. General data on the peculiarities of the hydrometeorological regime of the Caspian Sea are presented in [10,11]. A number of works are devoted to the sea level variability and storm surge phenomena, which are characteristic especially of the North Caspian [6,12–18].

The regime and extreme characteristics of wind waves were considered in a number of studies [19–22]. Wave climate of the Caspian Sea based on low-resolution NCEP/NCAR reanalysis is presented in [19,23–25]. The main features of the wave climate in the Caspian Sea for 2002–2013 based on altimetry data are displayed in [26]. Detailed estimates of monthly mean satellite-based wave heights for the Caspian Sea are given in [27]. The wave energy potential was estimated in [28–30]. Detailed wind wave hindcast is shown in [20,31], and the maximum wave height of 3% probability of exceedance was 11.7 m. Some studies presented the results of wind waves variability based on direct wave measurements [32] and wind variability based on Merra-2 reanalysis and wind measurements [33].

Some publications consider the hindcast and climate projections of the frequency of occurrence of synoptic conditions which cause severe hydrometeorological events including storm waves [34,35]. However, these papers also use wind data from the NCEP/NCAR reanalysis. The Atlantic and Pacific oceans' influence on meteorological parameters in the Caspian Sea area, including wind, is also noted in [14].

The variability of wind and wave regimes is also affected by climate warming and changes in the amount of ice in the Caspian Sea, as the number of mild and moderate winters has increased in the last decade [36]. According to studies, there will be no ice in the Middle Caspian Sea by 2035 and in the North Caspian Sea by 2055 (except for some points in the northeast) [37].

However, there is no detailed analysis of storm activity for the Caspian Sea.

Earlier, our scientific group received some basic information about the wind wave climate of the Caspian Sea [20,31]. Preliminary results about the number of storms in the whole Caspian Sea were presented in [20], but there is no detailed analysis for the different regions and seasons. Further, nothing has been published about the joint analysis of storm waves and wind speed climate variability.

In work [38], similar methods of analysis were used, but the Caspian Sea was not considered, and it was not a joint analysis of wind and storm wave trends. Work [39] describes basic approaches to the study of wind waves in seas and oceans and does not offer any methods to analyze storm activity.

On the other hand, there are many works devoted to detailed analysis of storm activity in the Atlantic Ocean, the Mediterranean, the Baltic Seas, and other seas based on different methodology [38,40–43].

The main goal of this research is to analyze the number of storms from 1979 to 2020 in different parts of the Caspian Sea and its connection to long-term wind variability. To assess the variability of wind and wave conditions in the Caspian Sea, we used a wave model and reanalysis data.

## 2. Data and Methods

The wave model WAVEWATCH III [44] was used to analyze wind waves in the Caspian Sea. High resolution NCEP/CFSR/CFSv2 reanalysis data from 1979 to 2020 were used as input wind data. The spatial resolution of the reanalysis is ~0.2–0.3° and the time step is 1 h. The ice fields were taken from the OSI-450 database and the NCEP/CFSv2 reanalysis. Calculations were performed on an unstructured triangulation grid. The grid spacing is about 10 km in the open part of the Caspian Sea and up to 900 m in the coastal zone, and the total number of nodes is 17,529 (Figure 1). Different colors indicate different parts of the Caspian Sea: the North Caspian (NC), the Middle Caspian (MC), and the South Caspian (SC). We used the method of dividing the Caspian Sea from [7–9]. The ST6 scheme [44] was used to generate waves in the model. A more detailed description of the model and its implementation for the Caspian Sea is given in [20,31].

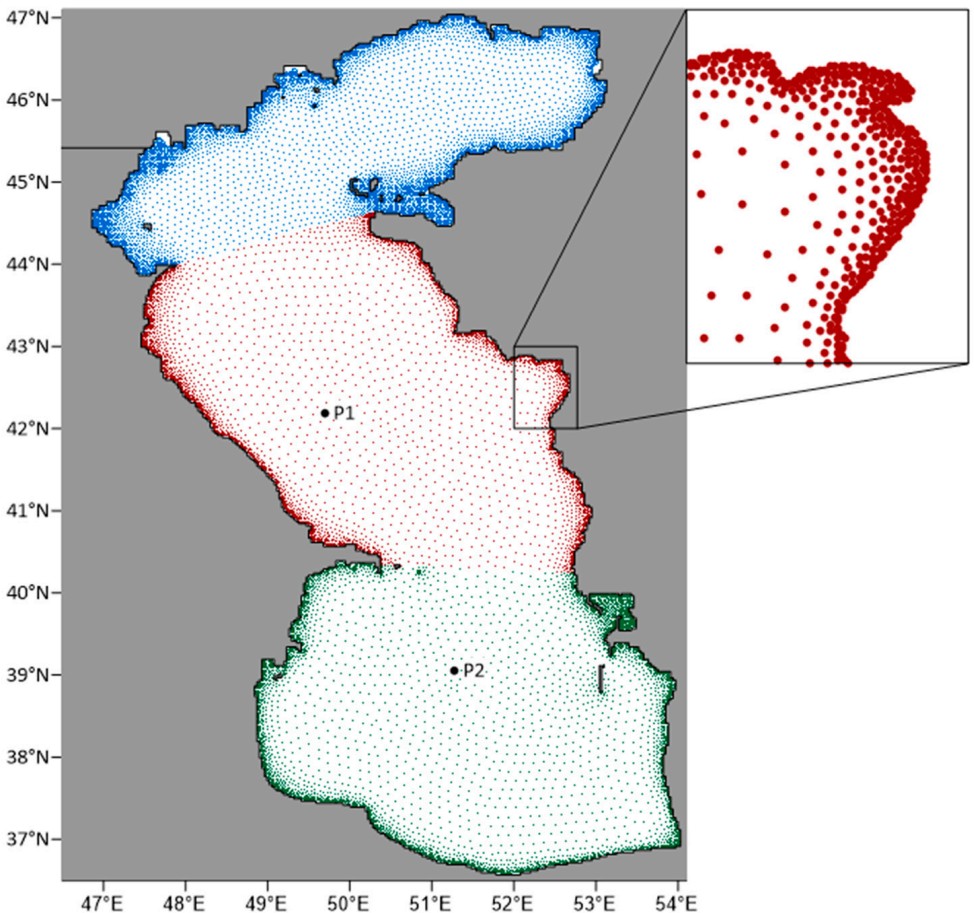

**Figure 1.** The computational unstructured grid for the WAVEWATCH III model (blue color—nodes in the North Caspian; red—in the Middle; green—in the South). P1 is a point in the Middle Caspian, and P2 is a point in the South Caspian.

Earlier [20,31], we evaluated the quality of model results. The results of comparison of model data with direct wave measurements and altimeter data (rads.tudelft.nl, accessed on 25 May 2023) for a point in the Middle Caspian are shown in Figure 2. The simulation quality may be visually assessed as satisfactory in Figure 2a. Unfortunately, we had only the image of wave measurements which was published in [32], and we could not calculate statistics. The correlation coefficient for the model and the altimeter data (Figure 2b) is 0.92 and the root mean square error is 0.28 m. These results for the wave model quality generally correspond to modern wave model implementations [38,39].

The model outputs include wind wave characteristics with a 3-h time step from 1979 to 2020 (42 years). Significant wave heights (SWH), wave direction, and wind components (U and V) were used in this research.

The Peak Over Threshold (POT) methodology was used to analyze storm activity in the Caspian Sea (Figure 3). The application of this methodology can be found in [20,38,45,46]. The calculation procedure included the following steps: A criterion is given (in our case, we chose criteria of wave height from 2 to 5 m and wind velocity of 5, 8, 10, 12, and 14 m/s); if at least one node in the investigated sea area has SWH exceeding the criteria, then such an event is attributed to the storm case. When it decreases below the criterion, the event ends. The duration of the storm is calculated as the difference between the end time and the start time of the event. Storms are counted both over the entire Caspian Sea water area and for the separately selected areas shown in Figure 1. If the time between two storms is less than 9 h, it is considered to be one long storm. Once the time between them exceeds 9 h, the storms are counted as two separate storms.

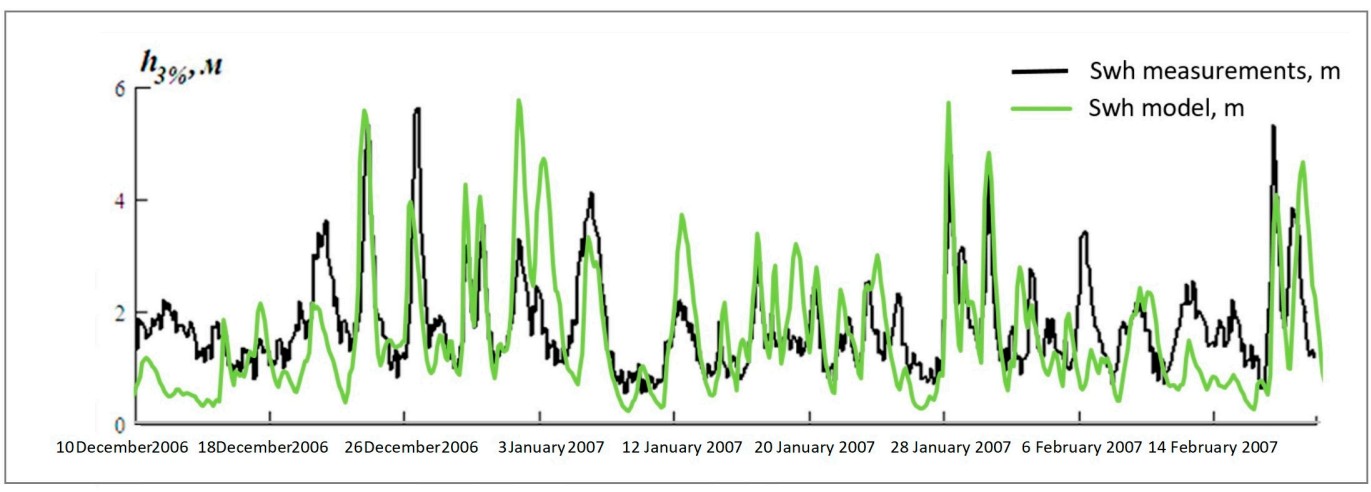

(a)

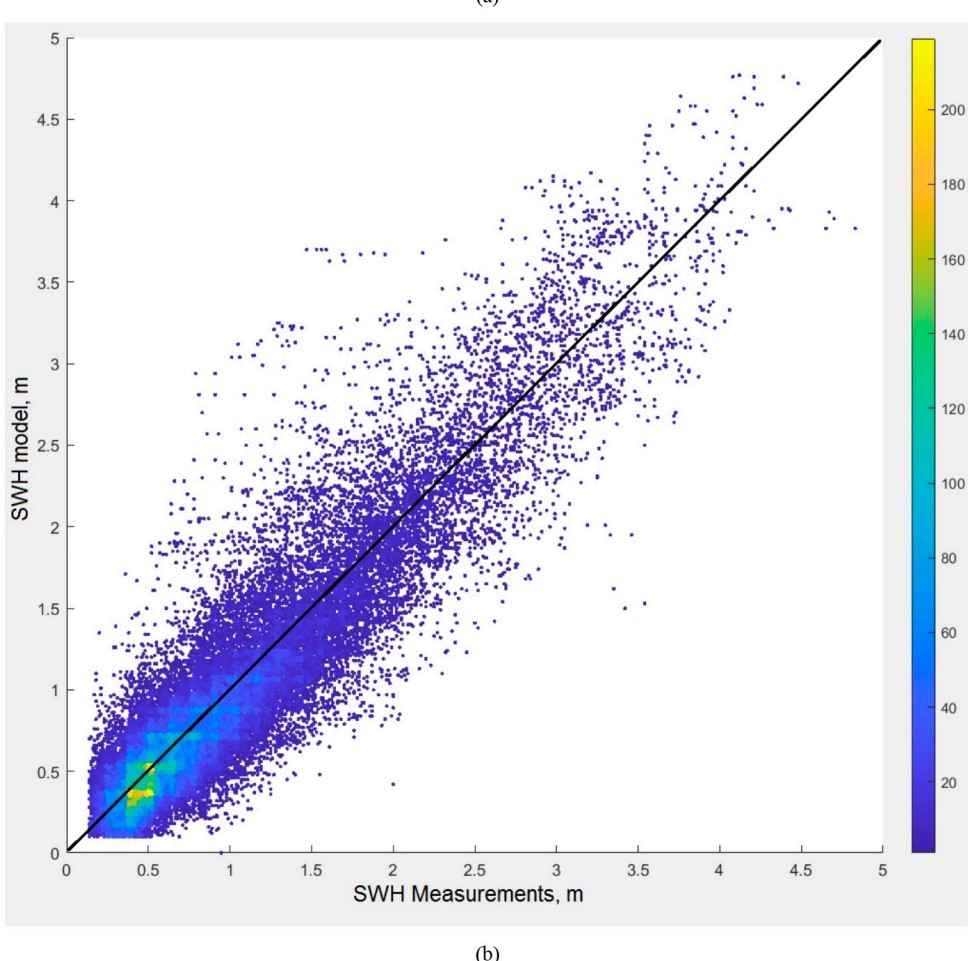

(b)

**Figure 2.** The wave height (3% probability of exceedance) according to measurement data at the point of the Middle Caspian Sea and the model results at this point (**a**), as well as the comparison of the SWH from satellite data and model results (**b**) [20].

The data obtained on the number of storms per year based on multiyear and seasonal multiyear sampling were examined for trends. A linear regression model ($y = a1x + a0$) was used to calculate the trends. The value of the trend is equal to the coefficient $a1$ of the linear trend and has the dimension of the characteristic $y$ per unit of discreteness. To assess the trends for significance, we analyzed the adequacy of regression models by Fisher's test and checked regression coefficients for significance. To assess adequacy, the hypothesis

of equality of variances H0: Dy = De and the alternative H1: Dy ≠ De were tested. Here, Dy is the variance of the model characterizing the variability of the regression line relative to the mean value of the model; De is the variance of the residuals characterizing the deviation of the regression equation from the actual values. The calculated Fisher's criterion was used for the estimation and compared with the critical value of the Fisher's criterion at a given significance level of 5%. If the empirical value of Fisher's criterion was greater than the critical value, the hypothesis of equality of variance was rejected, which in this case means the adequacy of the regression model or, in other words, the significance of the linear trend. To test the regression coefficients for significance, the hypothesis H0: $a = 0; b = 0$ with H1: $a \neq 0; b \neq 0$. For verification, Student's criteria were calculated.

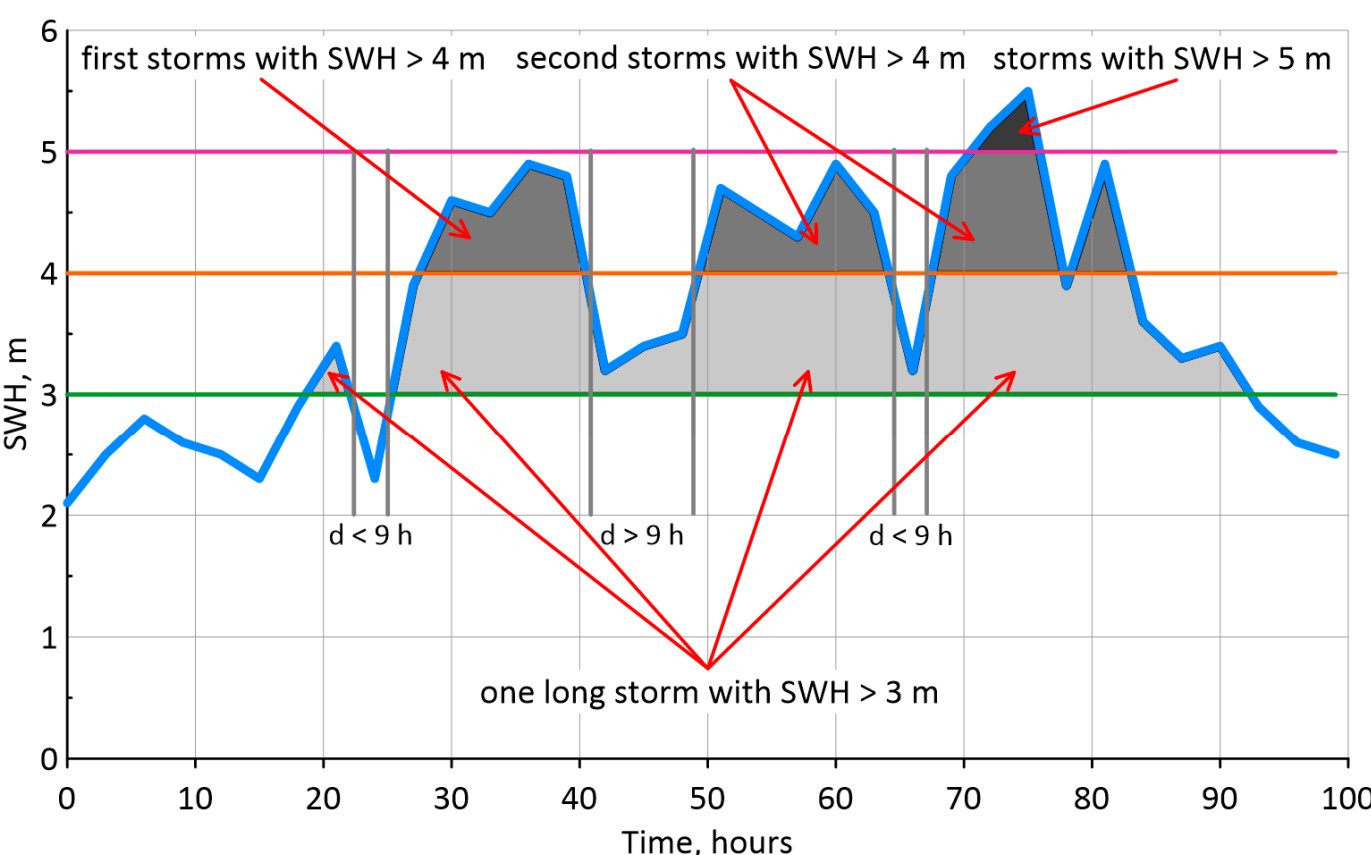

**Figure 3.** Visual representation of the application of the POT method for storm analysis.

## 3. Results and Discussion

### 3.1. Seasonal Variability of Significant Wave Heights

Based on the modeling data, mean and maximum multiyear values of SWH, and the 95th percentile, SWH distribution maps were obtained for the entire period and for standard calendar seasons: winter (December–February), spring (March–May), summer (June–August), and autumn (September–November).

Figure 4 shows the distribution of maximum (a), mean (b), and the 95th percentile SWH (c) over the entire period. The maximum SWH was located in the Middle Caspian and was 8.17 m. The maximum values in the North Caspian did not exceed 5 m. The local maximum in the South Caspian was ~7 m. The maximum mean SWH for the entire period was also observed in the MC and did not exceed 1.5 m. In the NC and SC, the mean SWH values for the entire period were less than 1 m. The maximum of the 95th percentile SWH for the entire period was observed in the MC and it was more than 2.5 m. The 95th percentile SWH was around 1.5–2.2 m in the SC and around 0.7–1 m in the NC.

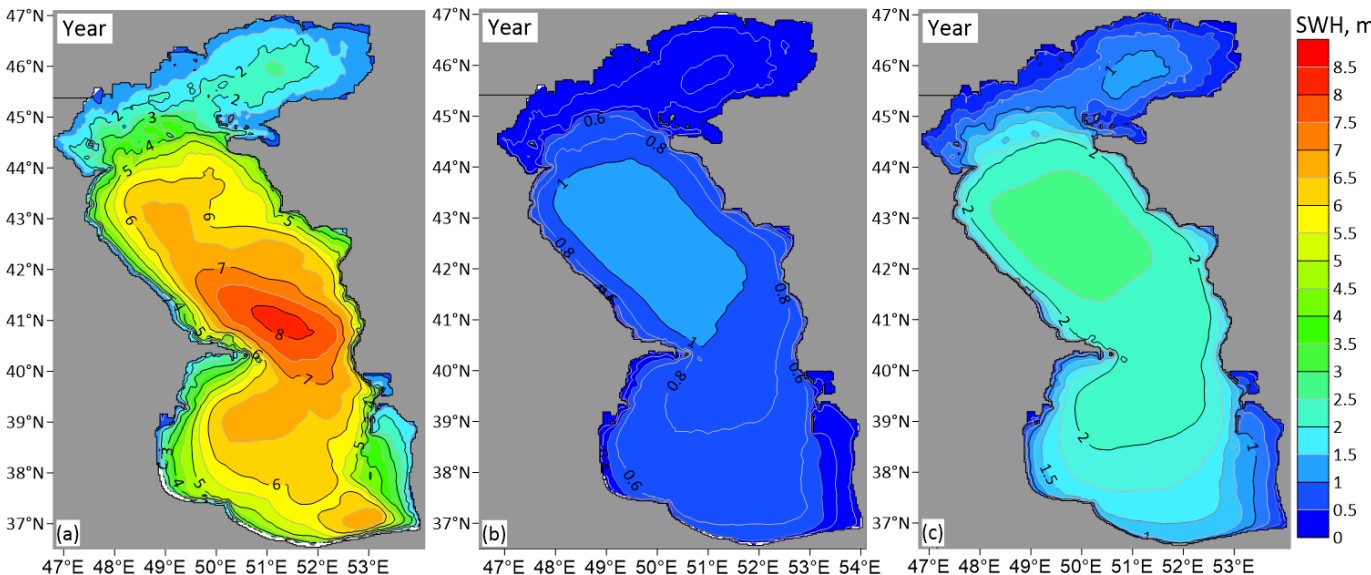

**Figure 4.** The long-term maximum (**a**), mean (**b**), and the 95th percentile (**c**) SWH for the period 1979 to 2020.

SWH of 6.9 and 5.6 were observed in spring (Figure 5) and summer (Figure 6), respectively. The SWH maxima were in the MC, close to the Apsheron Peninsula. Mean wave heights did not exceed 1.2 m in spring and 0.9 m in summer. It is worth noting that mean wave heights above 0.8 m in spring were more typical of the MC, while in summer, these values were located at the boundary between the MC and SC. The maximum of the 95th percentile SWH did not exceed 2.7 m in spring and 2.3 in summer.

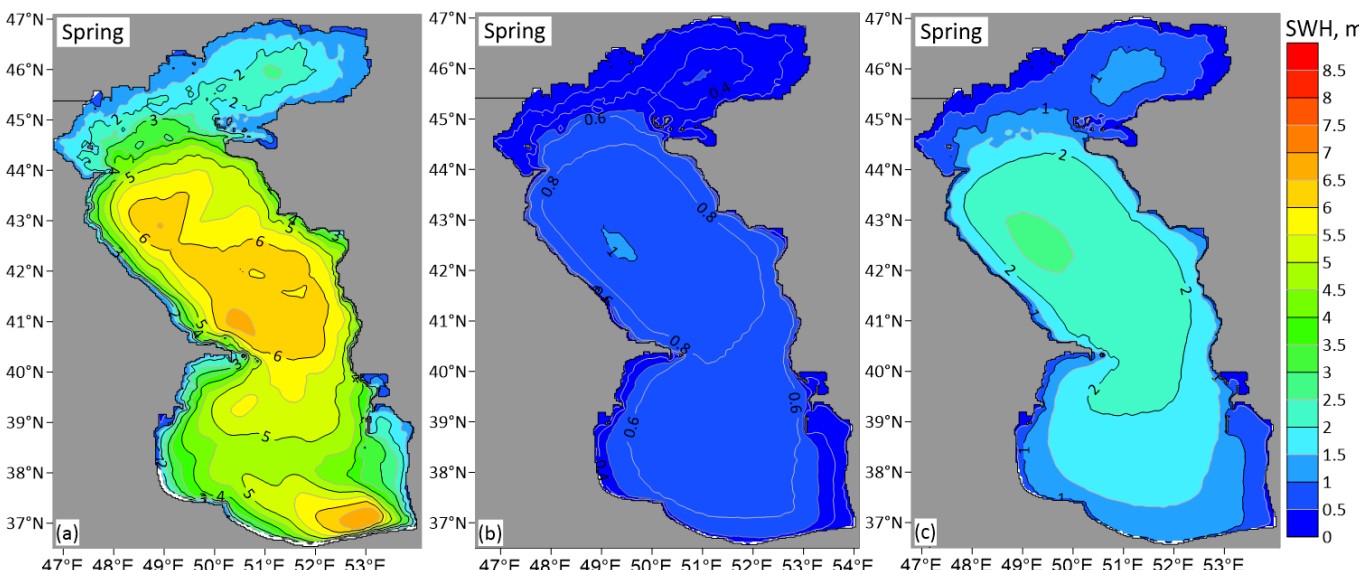

**Figure 5.** The long-term maximum (**a**), mean (**b**), and the 95th percentile (**c**) SWH for spring.

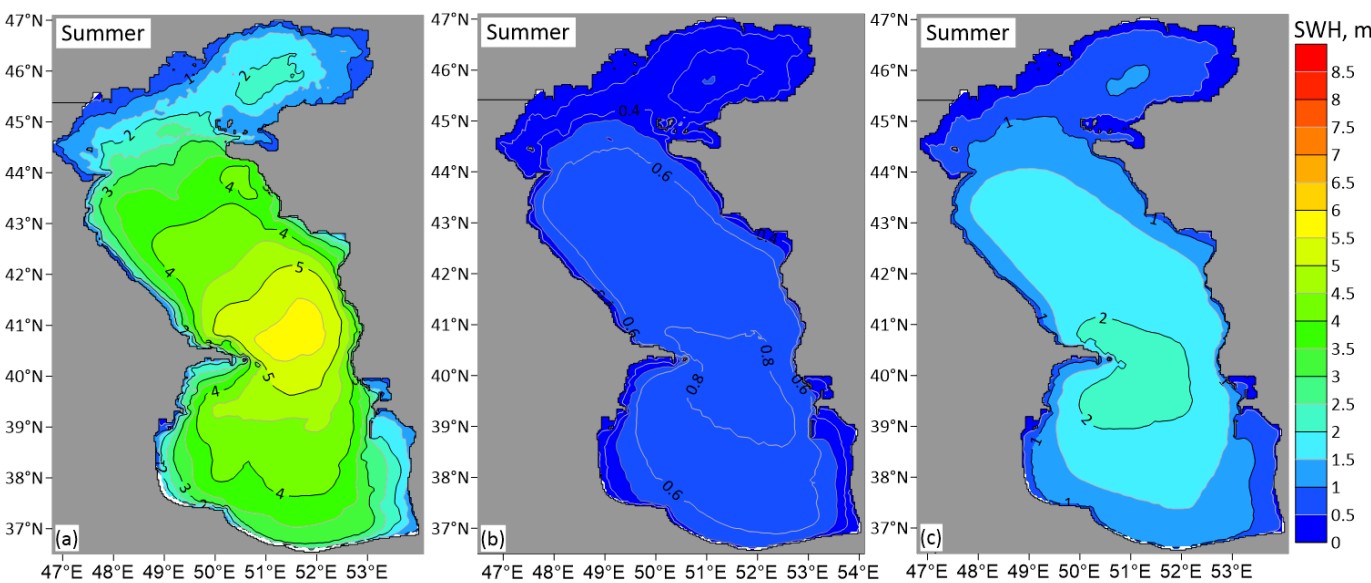

**Figure 6.** The long-term maximum (**a**), mean (**b**), and the 95th percentile (**c**) SWH for summer.

The autumn period (Figure 7) was characterized by three local SWH maxima—two in the MC (>6 m and >7 m) and one in the SC (>6 m). The absolute SWH maximum for this season was 7.18 m. The maximum mean values were also characteristic of the MC. The 95th percentile SWH greater than 2 m was characteristic of the SC, and greater than 2.5 m of the MC.

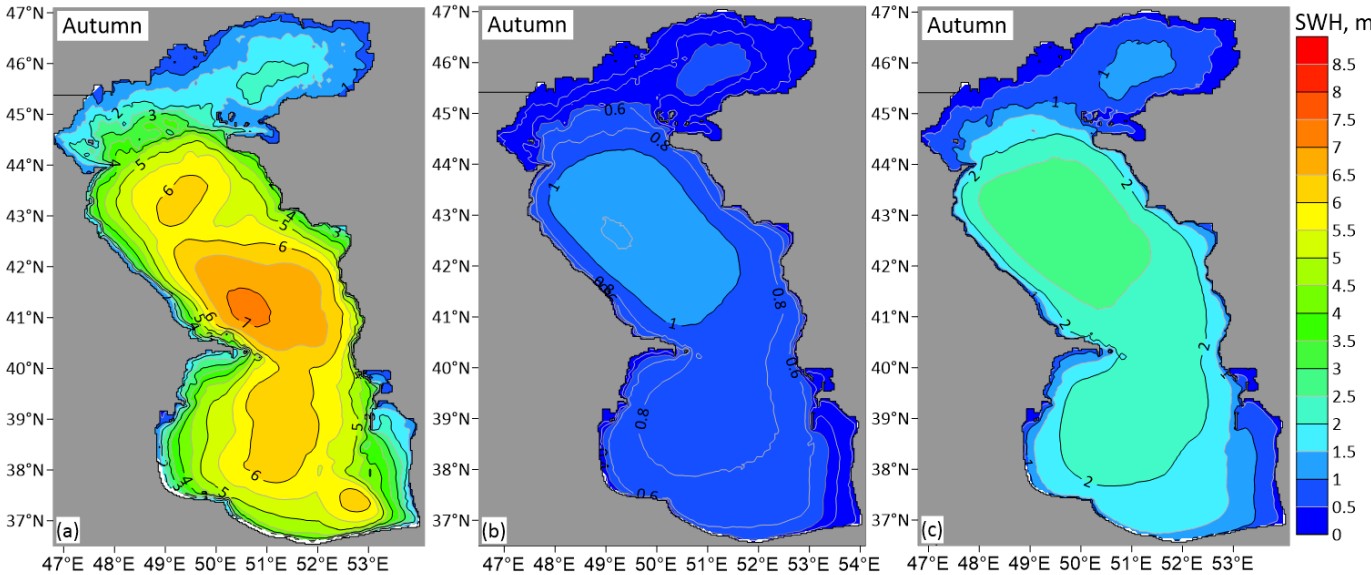

**Figure 7.** The long-term maximum (**a**), mean (**b**), and the 95th percentile (**c**) SWH for autumn.

The maximum SWH for the winter period was 8.17 in the MC (Figure 8). Mean SWH up to 1.5 m were also characteristic of MC. In the NC, mean SWH did not exceed 0.8 m, and for the SC, 1.2 m. The maximum SWH of the 95th percentile was 3.3 m. As can be seen from Figure 6, the winter period was characterized by the highest SWH by absolute maximum, mean values, and the 95th percentile. The same results were obtained in [20,31], but new maps include the 95th percentile analysis, and they significantly complement the knowledge about the wave climate of the Caspian Sea.

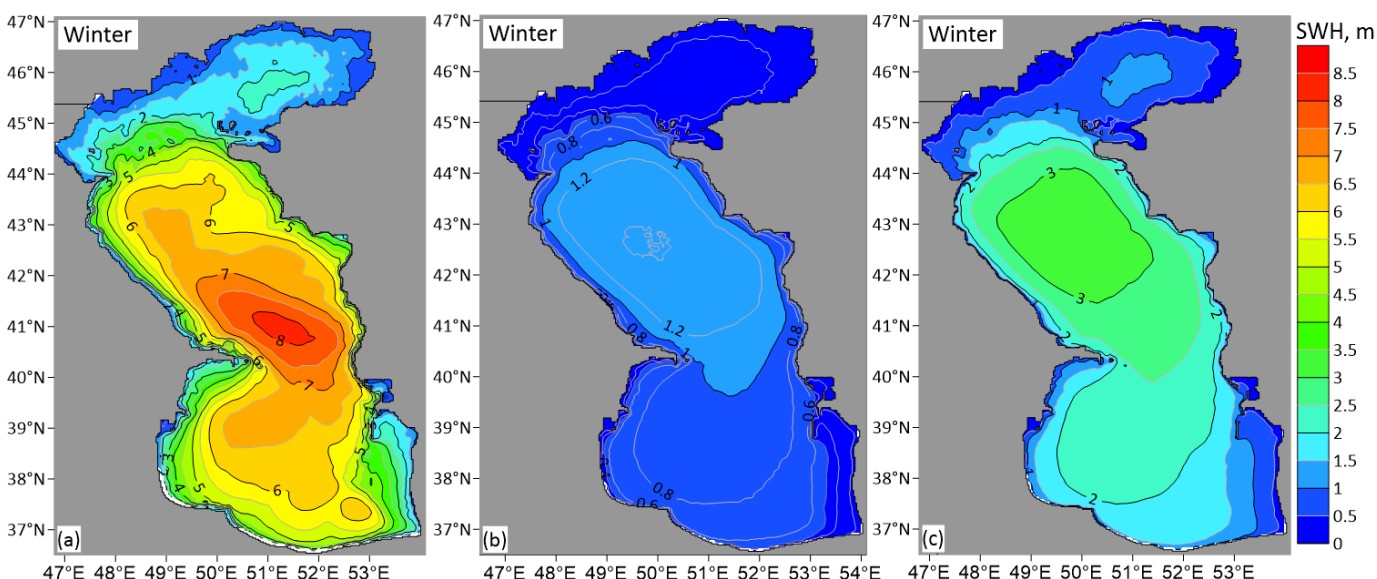

**Figure 8.** The long-term maximum (**a**), mean (**b**), and the 95th percentile (**c**) SWH for winter.

### 3.2. Interannual and Seasonal Variability of Storm Activity in the Caspian Sea

The results of storm activity variability were based on the POT method. We calculated the number of storm events per year with SWH thresholds between 2 and 5 m.

The multiyear mean number of storms with SWH > 2 m was 90 and with SWH > 3 m was 45 (Figure 9). The maximum number of storms with SWH > 2 m was observed in 1985, 2009, and 2018 (101 events), while those with SWH > 3 m occurred in 2014 and 2016 (59 storms), and 2015 (62 storms). There was also an absolute maximum of storms with SWH > 4 m in 2014 (23 storms). Significant positive trends were found for storms with thresholds of 3 and 4 m. The maximum trend value of 0.24 storms/year was observed for storms with SWH > 3 m; i.e., every ten years, the number of storms with SWH > 3 m increases by 2.

The analysis also included the total duration (in days) of storms with different criteria for each year. Figure 9b shows that the duration of storms with SWH > 2 m has increased. The mean number of stormy days with SWH > 2 m was ~114 days per year, with a maximum of ~145 days in 2016. The trend was positive and significant: every 10 years, the number of stormy days with SWH > 2 m increased by 4 days. In addition, a significant positive trend was found for the total duration of storms with SWH > 3 m. The trend value was 2 days/10 years. Trends for the number of stormy days for other SWH thresholds were not significant.

In [20], the number of storms was analyzed, but trends were not evaluated for significance. The total duration of storms was not analyzed and there was no analysis of seasonal variability. Next, the significance of trends for different seasons of the year was assessed.

In spring, the total number of storms was small (Figure 10a). The mean value for the number of storms with SWH > 2 m was 22 and 10 for SWH > 3 m. The maximum number of storms with SWH > 2 m was observed in 2018 (31 events), and for > 3 m, in 2014, 2017, and 2020 (17 storms). Positive significant trends were found for storms with SWH >3 and >4 m. The maximum trend value was 1 storm/10 years (>3m). In spring, a significant positive trend for total storm duration was observed only for storms with SWH > 3 m (Figure 10b), and it increased by 1 day every 10 years.

Summer was characterized by almost no storms with SWH > 5 m, and storms with SWH > 4 m occurred no more than twice a year (1984, 1992, 2003) (Figure 10c,d). The maximum number of storms with SWH > 2 m was 29 in 2020. The maximum total storm duration for SWH > 2 m was observed in 1984 (30.5 days), and for storms with SWH > 3 m, it was also in 1984 (5.25 days). There were no statistically significant trends for either

number of storms or total storm duration. It is worth noting that there was a negative trend for storm duration in this season.

In autumn (Figure 10e,f), we can see a strong year-to-year variability in both the number of storms and in total storm duration. The mean number of storms was 24 and 12 for storms with SWH > 2 m and SWH > 3 m, respectively. The maximum for storm number with SWH > 2 m was observed in 2009 (32 events) and for SWH > 3 m in 1989 (21 events). Trends for storms SWH > 2 m and SWH > 3 m were positive and not significant, and for SWH > 4 m and SWH > 5m, they were negative and not significant. The maximum of total storm duration was 42 days (for storms SWH > 2 m—in 1998 and 2007). There were no statistically significant trends.

Figure 10g shows that the number of storms with SWH > 2 m in winter had a negative trend, which was in contrast to storms with other thresholds. This can be explained by the fact that the total duration of these storms was increasing, implying that there were fewer storms but longer durations. There were no significant trends in the number of storms. Total storm durations increased over time in winter (Figure 10h). A positive significant trend was found for storm duration with SWH > 2 m: every 10 years, storms became 2 days longer.

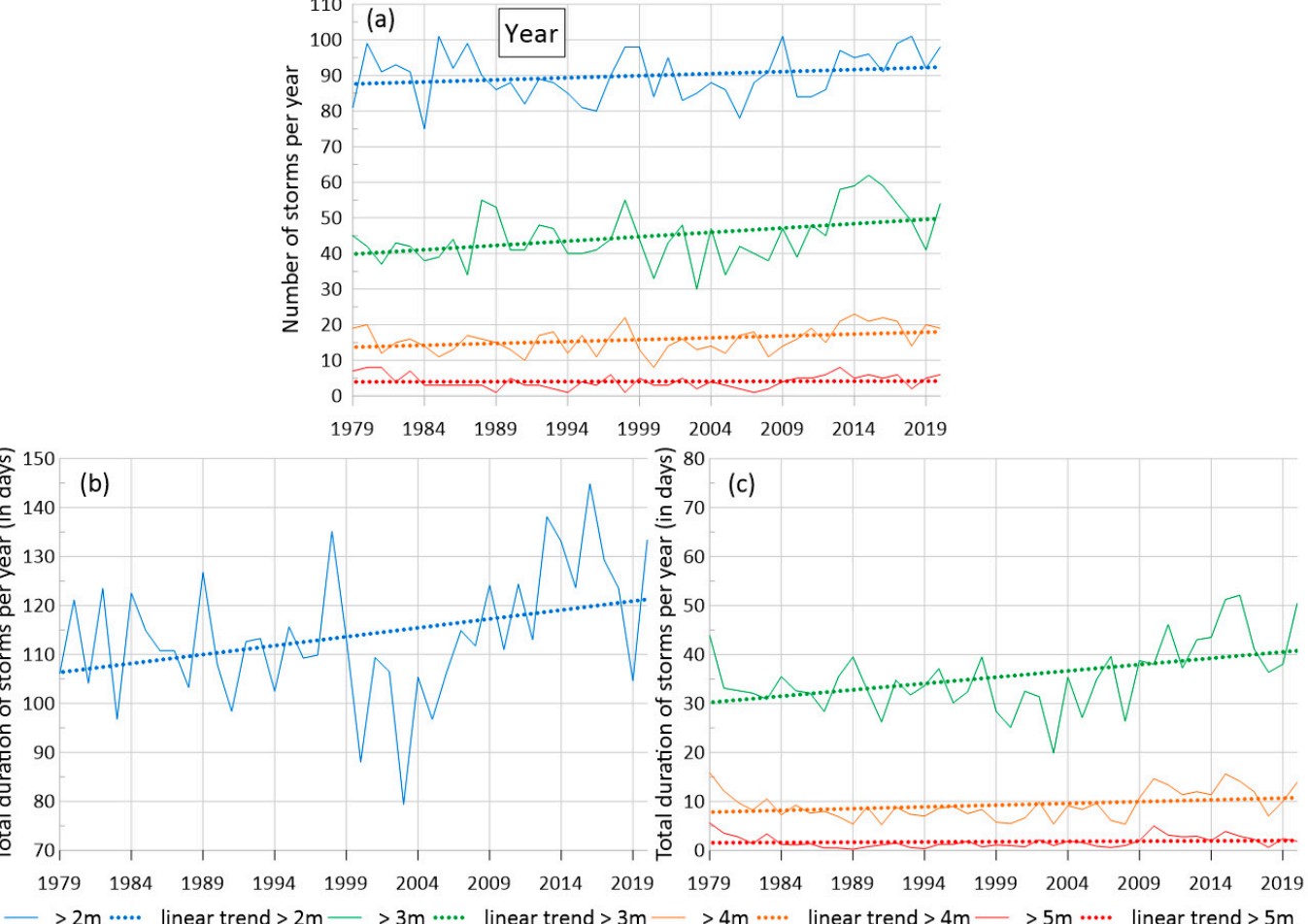

**Figure 9.** The number of storms with different SWH thresholds per year (**a**), total duration of storms (**b,c**), and their linear trends for 1979 to 2020.

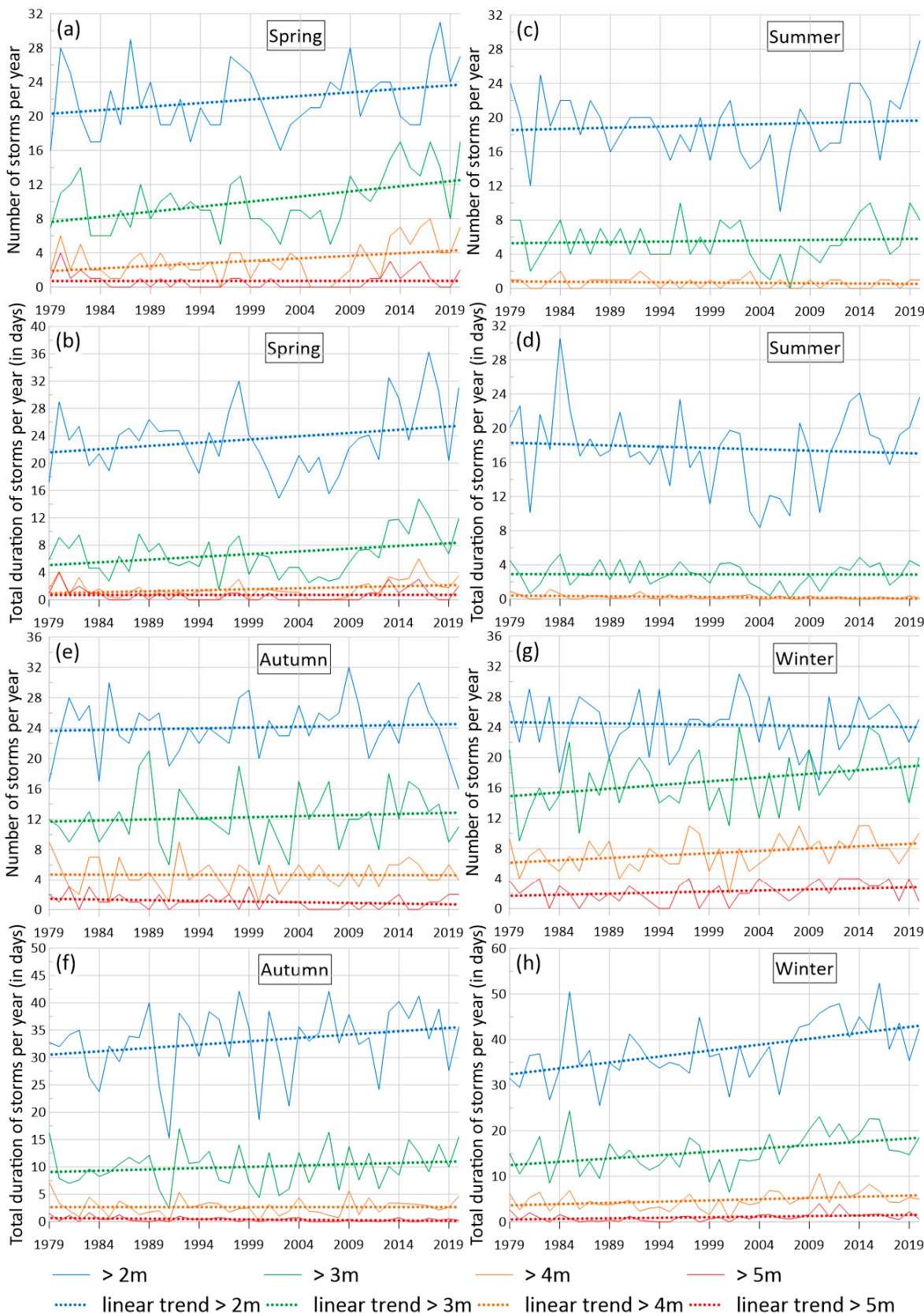

**Figure 10.** The number of storms with different SWH thresholds per year (**a**,**c**,**e**,**g**), total duration of storms (**b**,**d**,**f**,**h**), and their linear trends in spring, summer, autumn, and winter.

### 3.3. Interannual and Seasonal Variability of Storm Activity in Different Areas of the Caspian Sea

The number of storms and their total duration in different parts of the Caspian Sea has not been analyzed in previous studies. In the next stage of our research, we calculated the number of storms and their duration for the different parts of the Caspian Sea separately (see Figure 1).

For the North Caspian Sea, the changes in the number of storms were not large (Figure 11). The maximum trend value for the number of storms with SWH > 3 m was 1 storm/10 years. The trend was positive and significant. The total storm duration increased not only for storms with SWH > 3 m but also for SWH > 2 m. The maximum trend value was 2 days in 10 years and this trend was positive and significant. According to [47–49], the number of mild winters increased in the second half of the 20th century and early 21st century, which reduced the sea ice in the North Caspian Sea, which was associated with the presence of positive trends of number of storms and their duration in the NC.

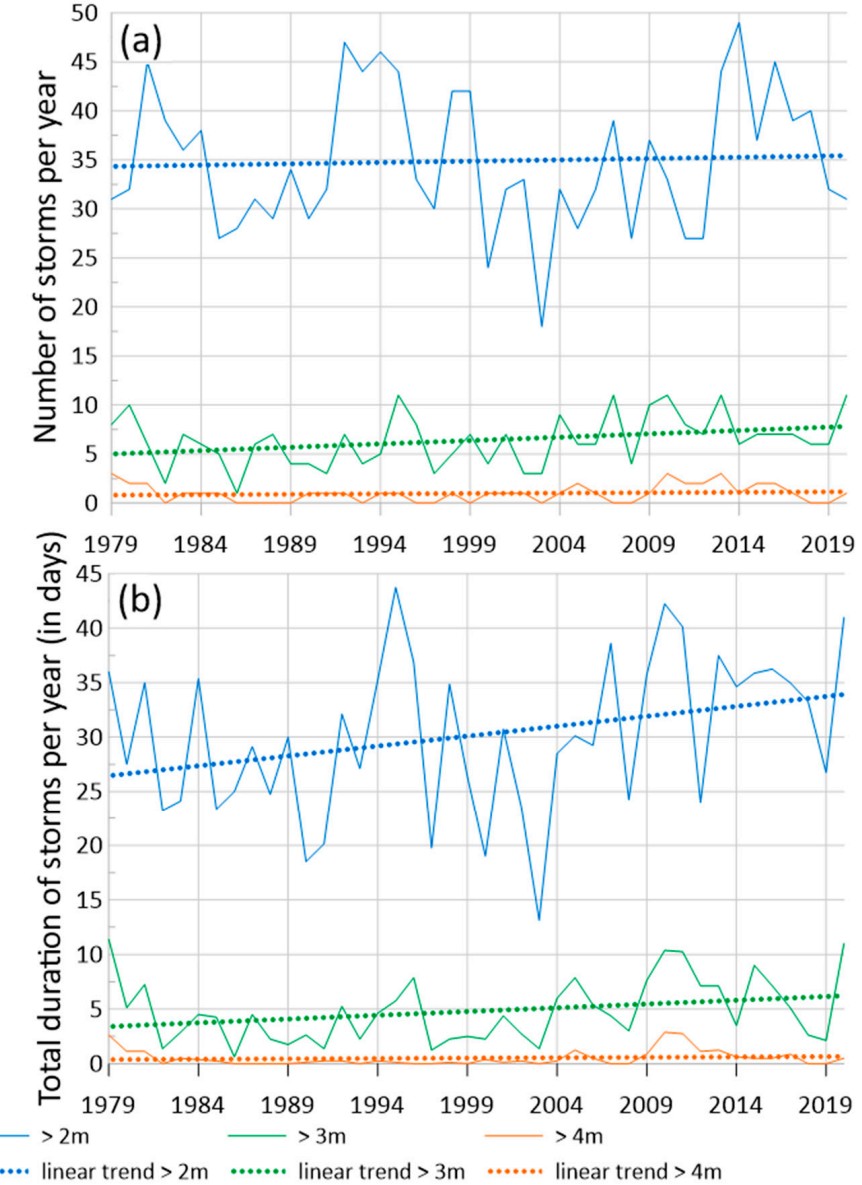

**Figure 11.** The number of storms with different SWH thresholds per year (**a**), the total duration of storms (**b**), and their linear trends in the North Caspian.

In the Middle Caspian, the mean number of storms with SWH > 2 m was 85 and 41 for SWH > 3 m (Figure 12). Significant positive trends were observed for storms with SWH > 3 m and > 4 m. The trend values were 3 storms/10 years and 1 storm/10 years, respectively. There was also one positive significant trend for total storm duration for storms with SWH > 3 m—2 days/10 years.

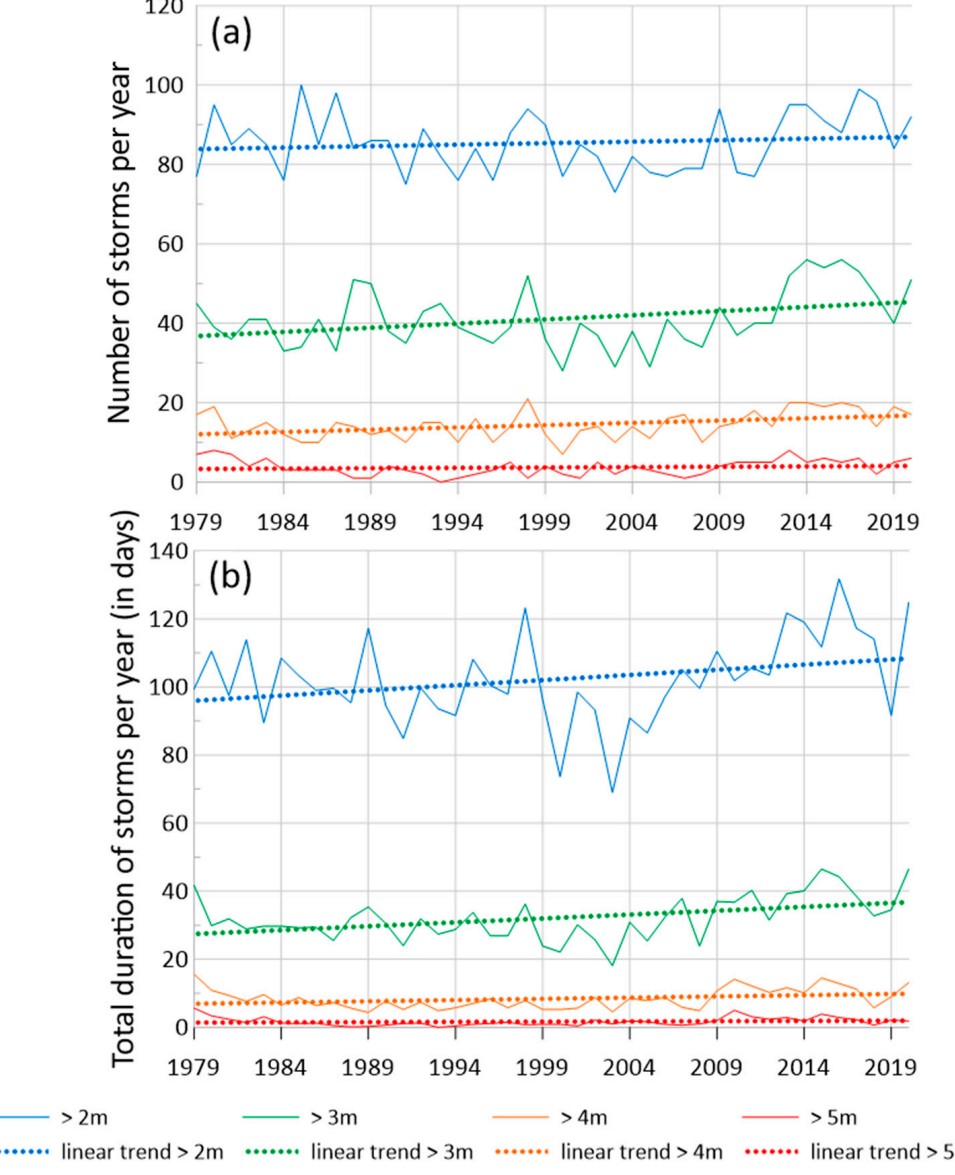

**Figure 12.** The number of storms with different SWH thresholds per year (**a**), the total duration of storms (**b**), and their linear trends in the Middle Caspian.

The mean number of storms in the South Caspian Sea for storms with SWH > 2 m and >3 m was 57 and 22 storms per year, respectively (Figure 13). The South Caspian Sea was characterized by negatively directed trends in the number of storms and the total duration of storms, but these trends were not significant.

The trends in the number of storms and the duration of storms in different areas of the Caspian Sea during different seasons of the year were also analyzed.

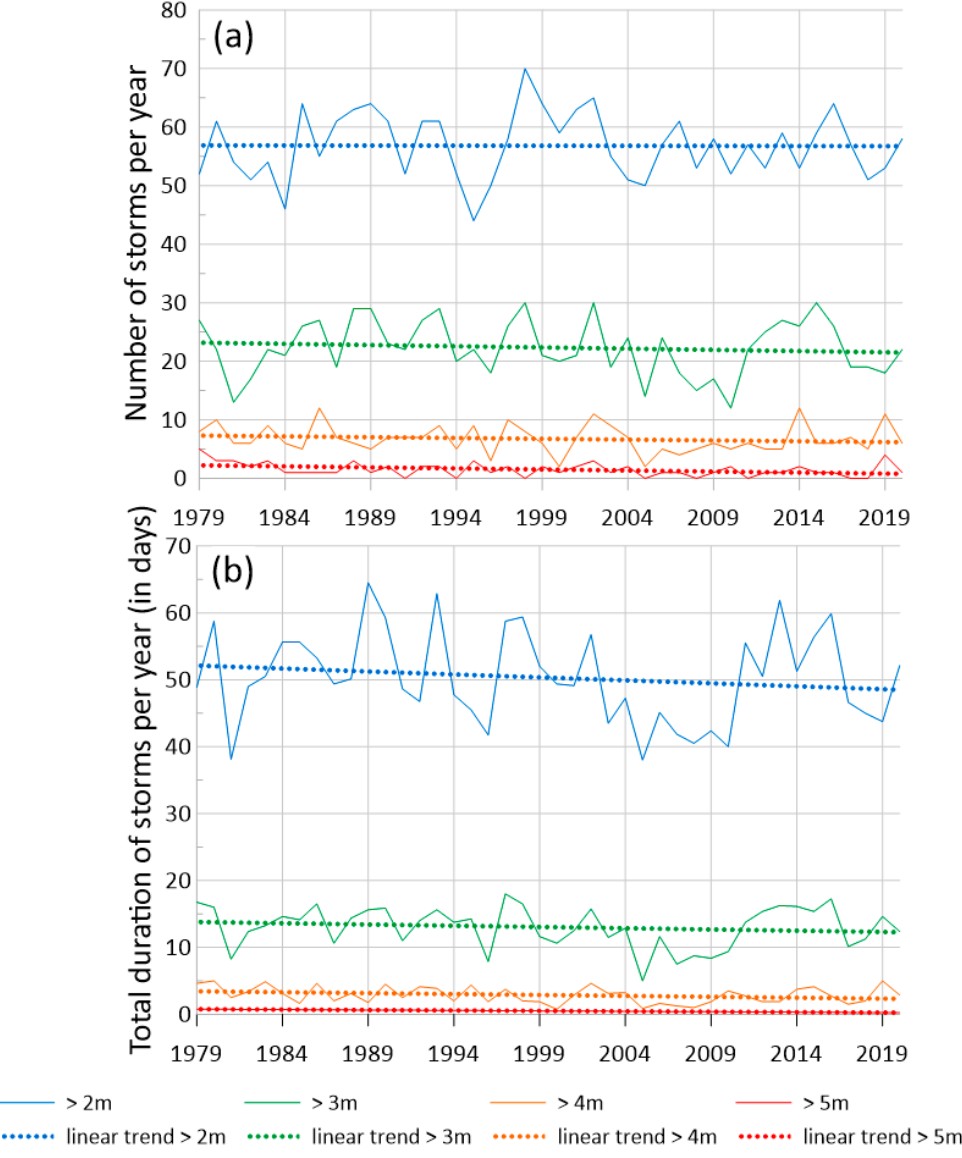

**Figure 13.** The number of storms with different SWH thresholds per year (**a**), the total duration of storms (**b**), and their linear trends in the South Caspian.

Figure 14 shows that in the North Caspian, the direction of trends in the number of storms and their total duration varied in different seasons of the year: positive trends were typical for autumn and winter, and negative trends for summer. In summer, storms with SWH > 4 m were not observed; in autumn and spring, storms of this criterion did not occur more than once a year. There were no statistically significant trends.

In the Middle Caspian (Figure 15), a decrease in the number of storms and their total duration in summer was found. For the other seasons (spring, autumn, and winter), positive trends were identified. The positive significant trends in the number of storms were found for spring (SWH > 3 and > 4 m). The maximum trend value of 1 storm/10 years for the number of storms with SWH > 3 m was found. For total storm duration, positive significant trends were observed for storms with SWH > 2 m and > 3 m.

The results of trend analysis for the South Caspian are presented in Figure 16. The positive trends were typical of the number of storms with SWH > 2 m and SWH > 3 m in spring and SWH > 2 m in winter. An increase in total storm duration was found for SWH > 2 m in spring. The other trends were negative, and there were no significant trends.

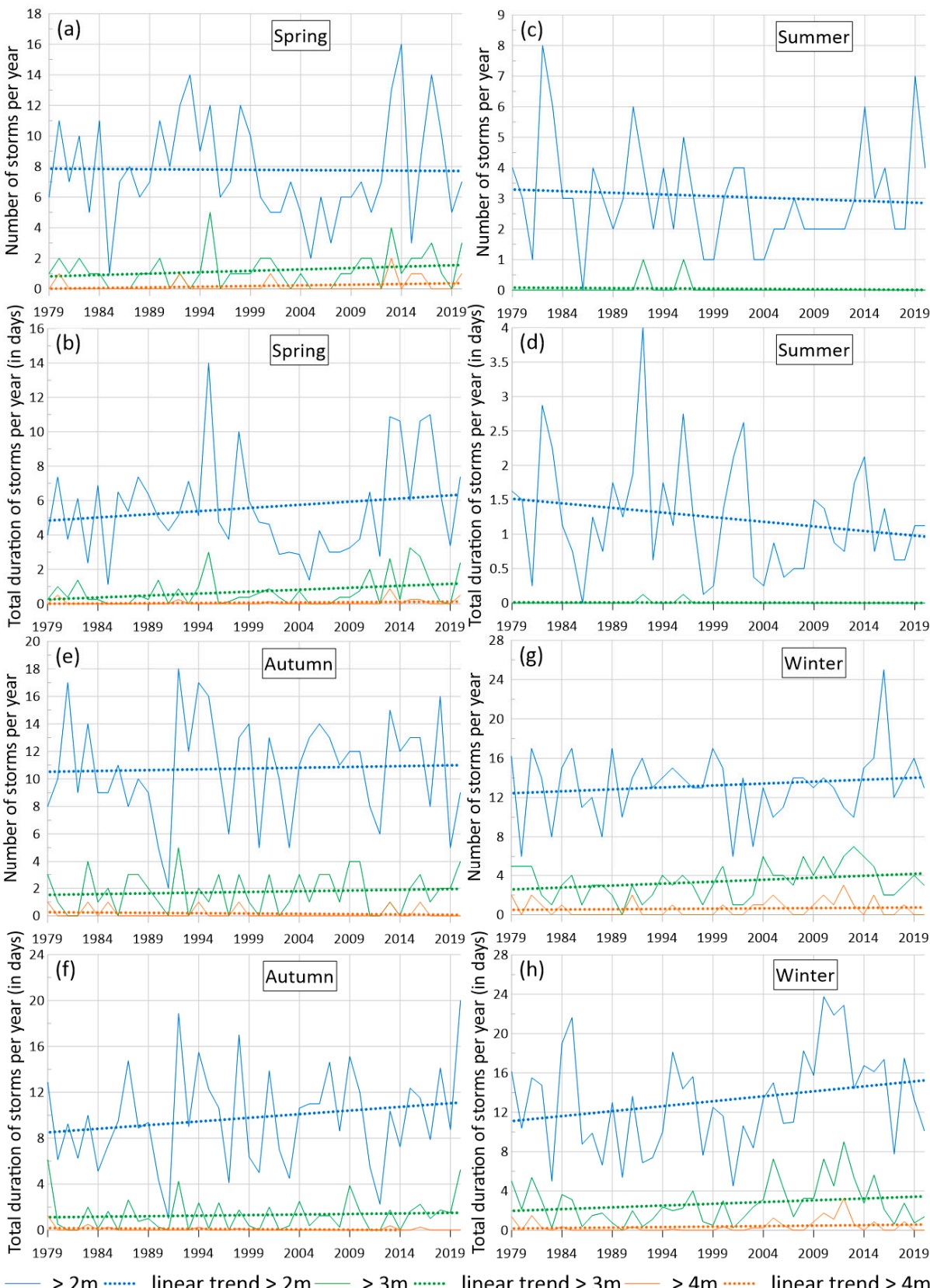

**Figure 14.** The number of storms with different SWH thresholds per year in spring, summer, autumn, and winter (**a**,**c**,**e**,**g**); the total duration of storms in spring, summer, autumn, and winter (**b**,**d**,**f**,**h**); and their linear trends in the North Caspian.

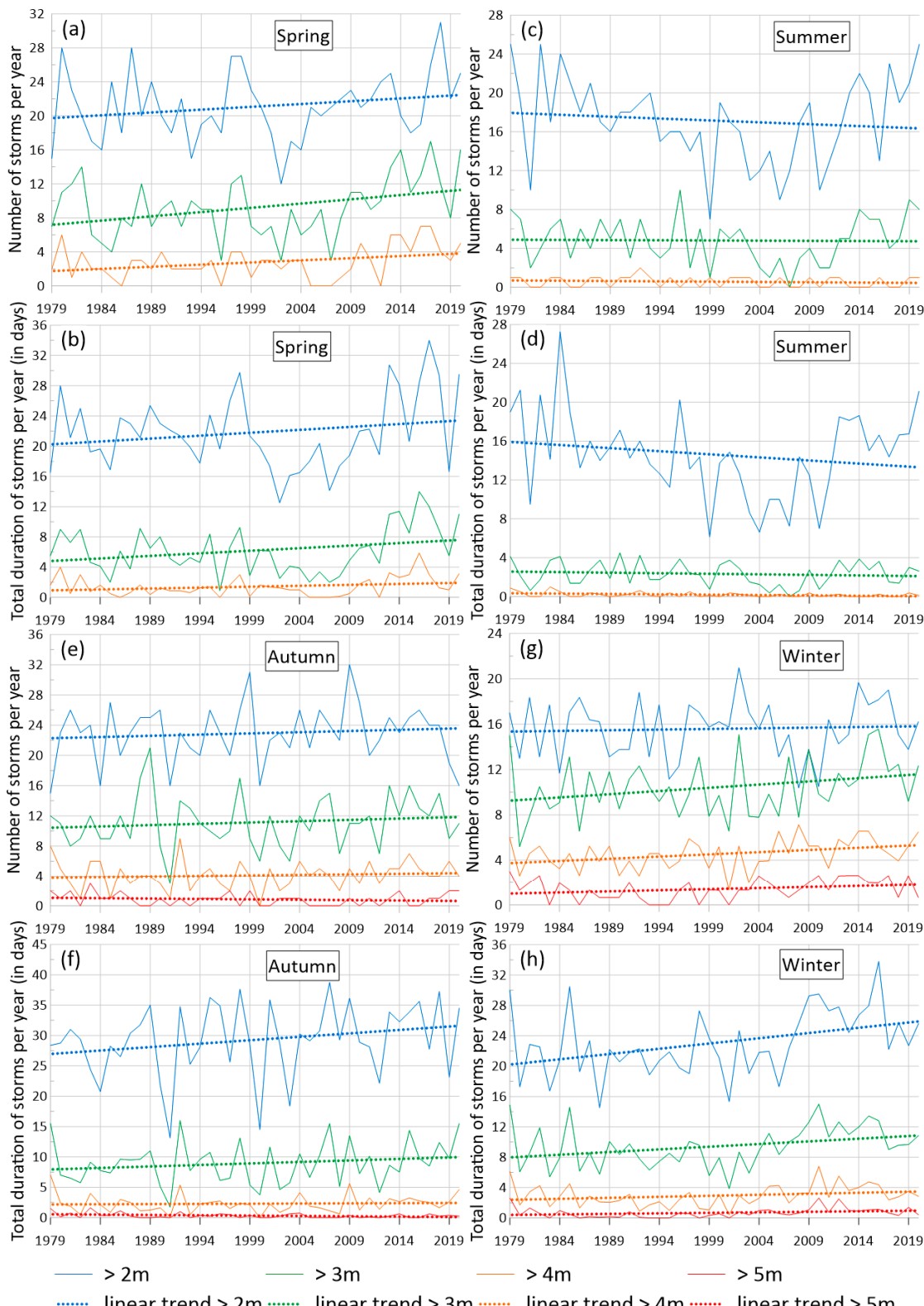

**Figure 15.** The number of storms with different SWH thresholds per year in spring, summer, autumn, and winter (**a**,**c**,**e**,**g**); the total duration of storms in spring, summer, autumn, and winter (**b**,**d**,**f**,**h**); and their linear trends in the Middle Caspian.

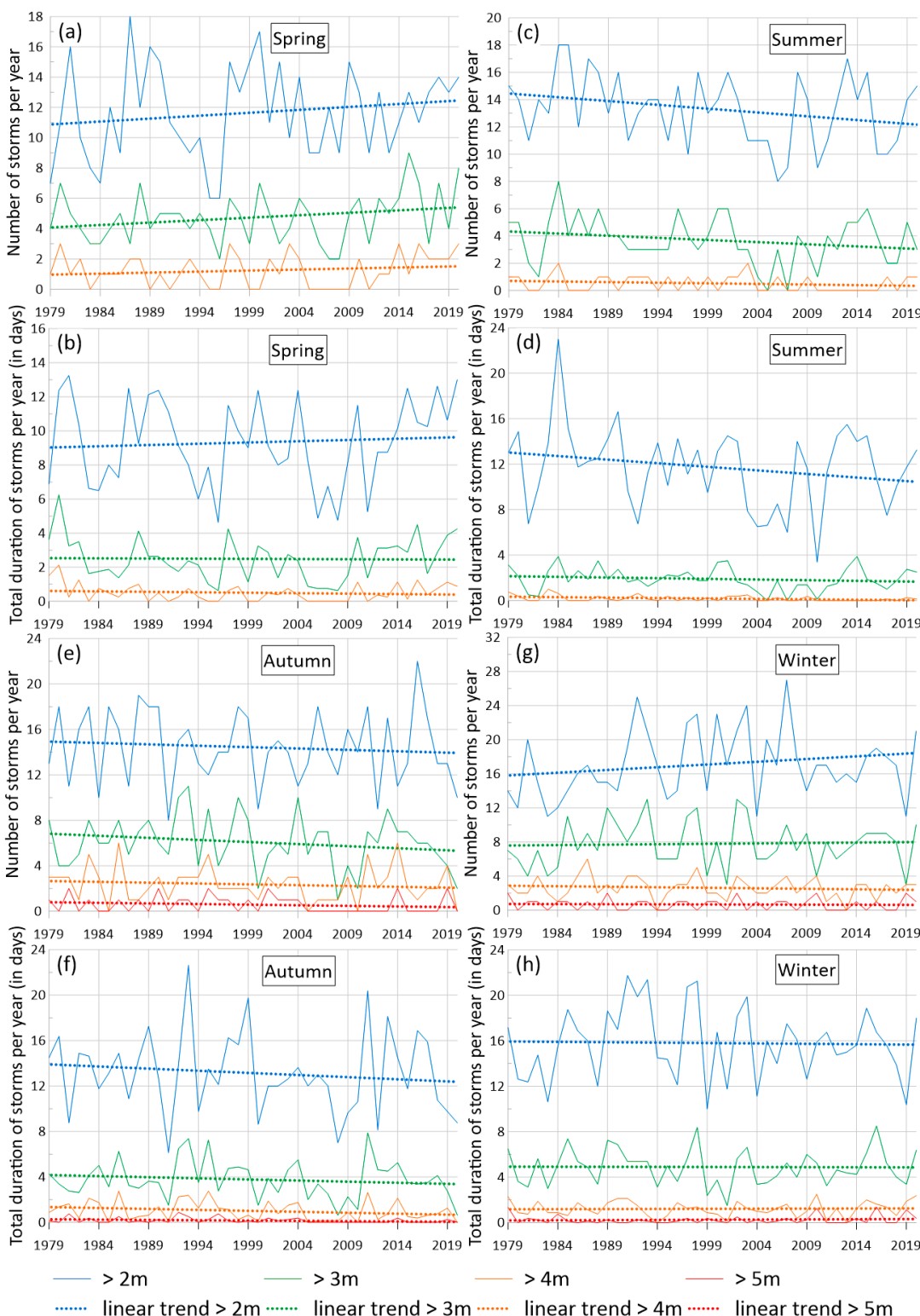

**Figure 16.** The number of storms with different SWH thresholds per year in spring, summer, autumn, and winter (**a**,**c**,**e**,**g**); the total duration of storms in spring, summer, autumn, and winter (**b**,**d**,**f**,**h**); and their linear trends in the South Caspian.

As a result of yearly data, it was found that significant positive trends in the number of storms were found only in the North and Middle Caspian (Figures 11–16 and Table 1). In the SC, a significant positive trend was also found in spring. Negative insignificant trends were found in the South Caspian. Trends in storm duration were also positive and

significant for the NC and MC (for the yearly data), and in winter in the Middle Caspian. Negative insignificant trends in storm duration were found in the South Caspian.

**Table 1.** Results of the trend analysis of number of storms and total storm duration for different parts of the Caspian Sea. Green cells mean that the empirical value of the coefficient is statistically significant and the trend is statistically significant; "-" means that trend analysis was not conducted. a1 is the trend value, and Tr_dir is the trend direction.

| | | | Number of Storms | | | | Total Duration of Storms | | | |
|---|---|---|---|---|---|---|---|---|---|---|
| | | | >2 | >3 | >4 | >5 | >2 | >3 | >4 | >5 |
| North Caspian | Year | Mean | 35 | 6 | 1 | - | 30.188 | 4.810 | 0.518 | - |
| | | a1 | 0.027 | 0.068 | 0.008 | - | 1.457 | 0.550 | 0.055 | - |
| | | Tr_dir | up | up | up | - | up | up | up | - |
| | Spring | Mean | 8 | 1 | 0 | - | 5.589 | 0.717 | 0.065 | - |
| | | a1 | −0.004 | 0.018 | 0.009 | - | 0.294 | 0.181 | 0.024 | - |
| | | Tr_dir | down | up | up | - | up | up | up | - |
| | Summer | Mean | 3 | - | - | - | 1.241 | - | - | - |
| | | a1 | −0.011 | - | - | - | −0.107 | - | - | - |
| | | Tr_dir | down | - | - | - | down | - | - | - |
| | Autumn | Mean | 11 | 2 | 0 | - | 9.810 | 1.307 | 0.071 | - |
| | | a1 | 0.012 | 0.010 | −0.005 | - | 0.508 | 0.077 | −0.039 | - |
| | | Tr_dir | up | up | down | - | up | up | down | - |
| | Winter | Mean | 13 | 3 | 1 | - | 13.173 | 2.711 | 0.381 | - |
| | | a1 | ~0 | ~0 | ~0 | - | 0.008 | 0.003 | 0.001 | - |
| | | Tr_dir | up | up | up | - | up | up | up | - |
| Middle Caspian | Year | Mean | 85 | 41 | 14 | 3.690 | 102.176 | 32.095 | 8.396 | 1.670 |
| | | a1 | 0.075 | 0.209 | 0.115 | 0.019 | 2.426 | 1.823 | 0.572 | 0.109 |
| | | Tr_dir | up | up | up | up | up | up | up | up |
| | Spring | Mean | 21 | 9 | 3 | 0.690 | 21.813 | 6.196 | 1.423 | 0.318 |
| | | a1 | 0.066 | 0.100 | 0.050 | −0.002 | 0.615 | 0.544 | 0.195 | 0.019 |
| | | Tr_dir | up | up | up | down | up | up | up | up |
| | Summer | Mean | 17 | 5 | 1 | 0.024 | 14.610 | 2.342 | - | - |
| | | a1 | −0.039 | −0.004 | −0.006 | −0.003 | −0.510 | −0.091 | - | - |
| | | Tr_dir | down | down | down | down | down | down | - | - |
| | Autumn | Mean | 23 | 11 | 4 | - | 29.274 | 8.952 | 2.298 | 0.330 |
| | | a1 | 0.032 | 0.035 | 0.014 | - | 0.904 | 0.395 | 0.048 | −0.087 |
| | | Tr_dir | up | up | up | - | up | up | up | up |
| | Winter | Mean | 16 | 10 | 5 | 1.394 | 23.048 | 9.418 | 2.929 | 0.679 |
| | | a1 | ~0 | 0.001 | ~0 | ~0 | 0.018 | 0.010 | 0.004 | 0.002 |
| | | Tr_dir | up | up | up | up | up | up | up | up |
| South Caspian | Year | Mean | 57 | 22 | 7 | - | 50.313 | 13.033 | 2.875 | - |
| | | a1 | −0.003 | −0.041 | −0.028 | - | −0.707 | −0.297 | −0.219 | - |
| | | Tr_dir | down | down | down | - | down | down | down | - |
| | Spring | Mean | 12 | 5 | 1 | - | 9.324 | 2.488 | 0.500 | - |
| | | a1 | 0.039 | 0.032 | 0.013 | - | 0.118 | −0.016 | −0.042 | - |
| | | Tr_dir | up | up | up | - | up | down | down | - |
| | Summer | Mean | 13 | 4 | 1 | - | 6.990 | 3.937 | 1.321 | - |
| | | a1 | −0.056 | −0.031 | −0.009 | - | −0.504 | −0.094 | −0.060 | - |
| | | Tr_dir | down | down | down | - | down | down | down | - |
| | Autumn | Mean | 14 | 6 | 2 | - | 13.134 | 3.765 | 1.003 | - |
| | | a1 | −0.024 | −0.037 | −0.015 | - | −0.300 | −0.156 | −0.134 | - |
| | | Tr_dir | down | down | down | - | down | down | down | - |
| | Winter | Mean | 17 | 8 | 3 | - | 15.801 | 4.892 | 1.212 | - |
| | | a1 | ~0 | ~0 | ~0 | - | −0.003 | −0.001 | ~0 | - |
| | | Tr_dir | up | down | down | - | down | down | down | - |

*3.4. Interannual and Seasonal Variability of Storm Activity and Wind Conditions at Different Points in the Caspian Sea*

The most interesting result from the above analysis was the different trend directions in the number and duration of storms in the Middle and South Caspian Sea (Table 1). For a more detailed analysis of the relationship between storm activity and wind variability, several points in the Middle and South Caspian Sea were selected (Figure 1, coordinates 49.6° E, 42.3° N and 39.06° E, 51.27° N, respectively). At these individual points, the number of storms and their duration were also calculated using the POT method.

Here, we provide the results of storm activity analysis based on individual point data. In the Middle Caspian, positive significant trends were observed for both the number of storms (SWH > 2–3 m) and total storm duration (SWH > 2–4 m). The maximum trend value for the number of storms was observed for storms with SWH> 2 m—16 storms/10 years. Since the mean number of storms with SWH > 2m was 58 at this point, it appears that the number of storms increased by 27% every ten years. The maximum trend value for total storm duration for storms with SWH > 3 m was 1 day/10 years.

Trends at the South Caspian Sea point were negative. The mean number of storms with SWH > 2 m was 35 storms per year, and for SWH > 3 m, it was 12. There were no statistically significant trends.

The trend analysis of the number of storms and storm duration was also carried out for the selected two locations during different seasons of the year (Tables 2 and 3).

**Table 2.** Correlation coefficients between the number of storms (at the point in the Middle Caspian and in the whole area of the Middle Caspian Sea) and the number of wind events. Correlation > 0.5 is marked in red.

| Wind Speed (m/s) | Duration (Hours) | Storms at the Point in the MC | | | | Storms in the MC | | | |
|---|---|---|---|---|---|---|---|---|---|
| | | >2 m | >3 m | >4 m | >5 m | >2 m | >3 m | >4 m | >5 m |
| >5 | 3 | −0.26 | −0.05 | −0.02 | 0.20 | −0.11 | −0.19 | −0.09 | 0.17 |
| | 6 | −0.24 | 0.02 | 0.07 | 0.21 | 0.01 | −0.16 | −0.06 | 0.18 |
| | 15 | −0.22 | 0.02 | 0.13 | 0.14 | 0.14 | −0.05 | −0.03 | 0.19 |
| | 24 | 0.02 | −0.01 | 0.19 | 0.13 | 0.32 | 0.04 | −0.05 | 0.34 |
| | 30 | 0.14 | −0.01 | 0.20 | 0.07 | 0.43 | 0.04 | −0.02 | 0.26 |
| >8 | 3 | 0.28 | 0.09 | 0.06 | −0.14 | 0.53 | 0.14 | 0.06 | 0.01 |
| | 6 | 0.36 | 0.14 | 0.00 | −0.27 | 0.66 | 0.19 | 0.09 | −0.11 |
| | 15 | 0.46 | 0.20 | 0.04 | −0.21 | 0.59 | 0.30 | 0.12 | −0.07 |
| | 24 | 0.46 | 0.30 | 0.23 | −0.07 | 0.58 | 0.39 | 0.32 | 0.05 |
| | 30 | 0.35 | 0.19 | 0.20 | −0.02 | 0.37 | 0.26 | 0.21 | 0.10 |
| >10 | 3 | 0.65 | 0.35 | 0.19 | −0.11 | 0.69 | 0.50 | 0.34 | −0.02 |
| | 6 | 0.69 | 0.39 | 0.26 | −0.11 | 0.71 | 0.49 | 0.34 | 0.05 |
| | 15 | 0.78 | 0.56 | 0.47 | 0.14 | 0.65 | 0.54 | 0.56 | 0.22 |
| | 24 | 0.68 | 0.42 | 0.47 | 0.26 | 0.48 | 0.46 | 0.51 | 0.34 |
| | 30 | 0.55 | 0.24 | 0.25 | 0.11 | 0.34 | 0.35 | 0.36 | 0.16 |
| >12 | 3 | 0.86 | 0.56 | 0.42 | 0.16 | 0.59 | 0.70 | 0.52 | 0.21 |
| | 6 | 0.85 | 0.66 | 0.49 | 0.18 | 0.58 | 0.78 | 0.61 | 0.25 |
| | 15 | 0.67 | 0.67 | 0.66 | 0.49 | 0.36 | 0.69 | 0.73 | 0.50 |
| | 24 | 0.39 | 0.48 | 0.48 | 0.30 | 0.27 | 0.43 | 0.43 | 0.37 |
| | 30 | 0.33 | 0.51 | 0.41 | 0.24 | 0.23 | 0.41 | 0.44 | 0.29 |
| >14 | 3 | 0.65 | 0.85 | 0.57 | 0.30 | 0.38 | 0.79 | 0.75 | 0.31 |
| | 6 | 0.61 | 0.82 | 0.66 | 0.40 | 0.49 | 0.72 | 0.70 | 0.51 |
| | 15 | 0.39 | 0.65 | 0.52 | 0.44 | 0.32 | 0.53 | 0.60 | 0.44 |
| | 24 | 0.34 | 0.59 | 0.63 | 0.44 | 0.35 | 0.52 | 0.61 | 0.49 |
| | 30 | 0.21 | 0.48 | 0.50 | 0.32 | 0.19 | 0.43 | 0.49 | 0.47 |

**Table 3.** Correlation coefficients between the number of storms (at the point in the South Caspian and in the whole area of the South Caspian Sea) and the number of wind events. Correlation > 0.5 is marked in red.

| Wind Speed (m/s) | Duration (Hours) | Storms at the Point in the SC | | | | Storms in the SC | | | |
|---|---|---|---|---|---|---|---|---|---|
| | | >2 m | >3 m | >4 m | >5 m | >2 m | >3 m | >4 m | >5 m |
| >5 | 3 | 0.23 | 0.02 | 0.10 | 0.18 | 0.37 | 0.22 | 0.26 | 0.23 |
| | 6 | 0.15 | 0.05 | 0.08 | 0.24 | 0.39 | 0.13 | 0.25 | 0.30 |
| | 15 | 0.14 | −0.07 | −0.15 | 0.33 | 0.37 | −0.10 | 0.03 | 0.04 |
| | 24 | 0.16 | 0.01 | −0.07 | 0.26 | 0.17 | −0.01 | 0.18 | 0.11 |
| | 30 | 0.25 | −0.05 | −0.04 | 0.31 | 0.16 | −0.06 | 0.15 | 0.08 |
| >8 | 3 | 0.21 | −0.01 | −0.42 | −0.20 | 0.52 | 0.00 | −0.15 | −0.32 |
| | 6 | 0.24 | −0.01 | −0.47 | −0.21 | 0.53 | 0.04 | −0.18 | −0.44 |
| | 15 | 0.46 | 0.11 | −0.43 | −0.17 | 0.47 | 0.24 | −0.16 | −0.38 |
| | 24 | 0.37 | 0.05 | −0.44 | −0.16 | 0.35 | 0.18 | −0.09 | −0.31 |
| | 30 | 0.29 | 0.02 | −0.47 | −0.06 | 0.25 | 0.22 | −0.15 | −0.31 |
| >10 | 3 | 0.74 | 0.46 | −0.13 | −0.20 | 0.68 | 0.51 | 0.29 | −0.02 |
| | 6 | 0.62 | 0.50 | −0.14 | −0.12 | 0.60 | 0.50 | 0.27 | 0.04 |
| | 15 | 0.46 | 0.35 | −0.15 | −0.11 | 0.35 | 0.40 | −0.02 | −0.10 |
| | 24 | 0.26 | 0.29 | −0.08 | 0.02 | 0.05 | 0.38 | 0.01 | −0.10 |
| | 30 | 0.21 | 0.32 | 0.19 | 0.23 | −0.06 | 0.26 | 0.30 | 0.07 |
| >12 | 3 | 0.67 | 0.78 | 0.07 | −0.12 | 0.35 | 0.64 | 0.28 | 0.16 |
| | 6 | 0.62 | 0.80 | 0.11 | −0.11 | 0.31 | 0.62 | 0.29 | 0.21 |
| | 15 | 0.41 | 0.64 | 0.13 | 0.10 | 0.08 | 0.50 | 0.28 | 0.19 |
| | 24 | 0.19 | 0.31 | 0.26 | 0.14 | −0.02 | 0.33 | 0.21 | 0.14 |
| | 30 | −0.11 | 0.15 | 0.34 | 0.21 | −0.26 | 0.20 | 0.31 | 0.12 |
| >14 | 3 | 0.51 | 0.80 | 0.39 | −0.04 | 0.16 | 0.52 | 0.55 | 0.25 |
| | 6 | 0.21 | 0.53 | 0.55 | 0.07 | −0.04 | 0.41 | 0.44 | 0.52 |
| | 15 | 0.26 | 0.44 | 0.31 | 0.22 | 0.04 | 0.34 | 0.27 | 0.30 |
| | 24 | −0.11 | −0.07 | 0.28 | 0.30 | −0.02 | 0.05 | 0.21 | 0.26 |
| | 30 | −0.10 | −0.03 | 0.19 | 0.19 | 0.16 | 0.15 | 0.10 | 0.28 |

For the MC point, positive significant trends were found for the number of storms with SWH > 3 m (winter period) and for total storm duration in spring (SWH > 3 m) and winter (SWH > 2 m and SWH > 3 m). At the South Caspian point, there was a positive trend for storms with SWH > 2 m, and a negative significant trend for storms > 4 m in summer and with SWH > 3 m in winter. The significant trend in total storm duration was also negative for storms with SWH > 4 m in summer.

The selected points reflect well the changes in storm activity in the areas—the Middle Caspian is characterized by positive significant trends, while the SC is characterized by negative trends.

Next, we moved on to analyze the variability of wind parameters in order to explain the detected trends in the number of storms.

Wind events with different wind speeds (>5, >8, >10, >12, and >14 m/s) of different durations (from 6 h to 30 h) were analyzed. The annual number of wind events of each wind speed and different duration were obtained. Next, the correlation analysis for storms and specific wind events was performed in the selected points and in the different parts of the Caspian Sea area. The correlation coefficients are presented below (Tables 2 and 3).

The maximum value of correlation coefficient for storms with SWH > 2 m in the Middle Caspian Sea of 0.86 was observed with wind events > 12 m/s and duration 6 h. Further, the correlation coefficient of 0.85 was found for storms with SWH > 3 m, wind events 14 m/s, and duration 6 h (Table 2). The maximum correlations for storms in the entire Middle Caspian Sea were 0.79 for storms with SWH > 3 m, winds > 14 m/s, and

duration 6 h and 0.78 for storms with SWH > 3 m, wind events > 12 m/s, and duration 9 h. A visual comparison of the graphs with the highest correlation is presented in Figure 17.

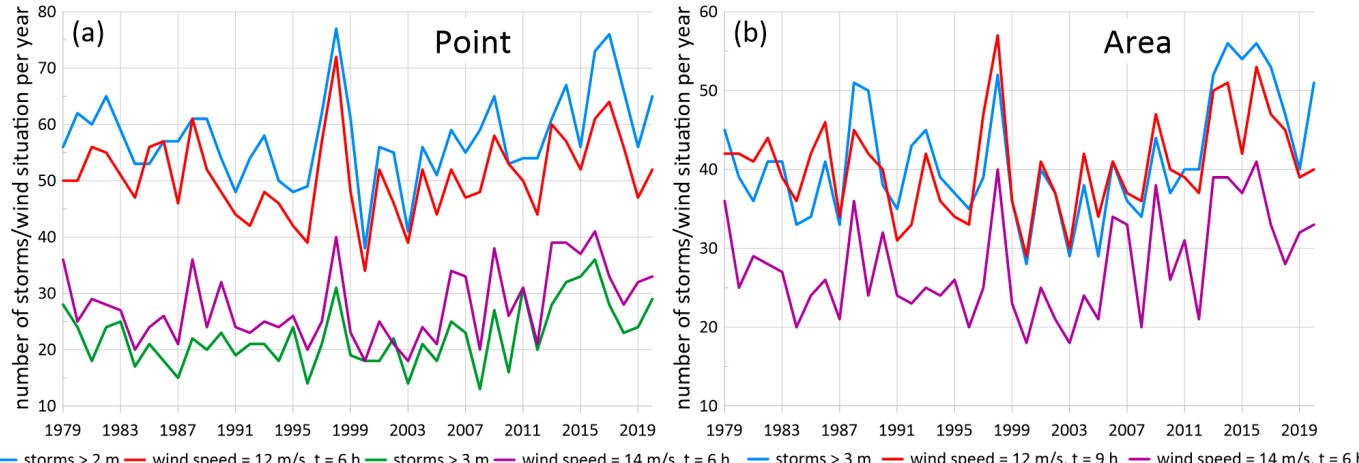

**Figure 17.** The number of storms at the point in the Middle Caspian Sea (**a**), the number of storms in the entire Middle Caspian (**b**), and their interrelation to wind situations.

The correlation coefficients for the South Caspian Sea are slightly worse than for the Middle Caspian Sea (Table 3). The maximum correlation for storms with SWH > 3 m in the South Caspian point was 0.80 for wind events > 14 m/s and duration of 6 h and 0.80 for wind events > 12 m/s and duration of 9 h. The maximum correlation for storm number in the SC area did not exceed 0.7. The highest correlation was observed here for storms with SWH > 2 m, wind events > 10 m/s, and duration of 6 h. A visual comparison of the graphs with the highest correlation is presented in Figure 18.

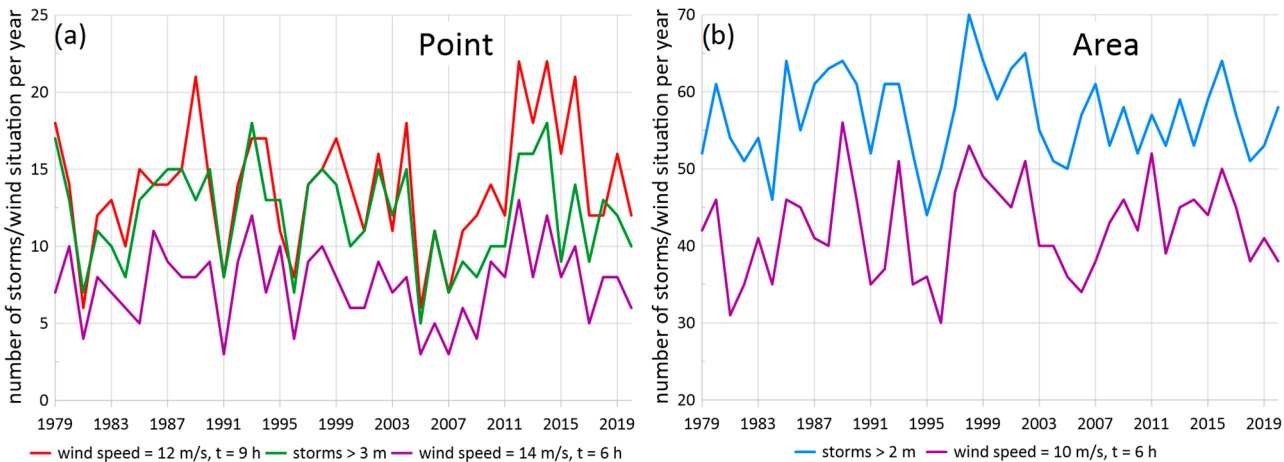

**Figure 18.** The number of storms at the point in the South Caspian Sea (**a**), the number of storms in the entire South Caspian Sea (**b**), and their interrelation to wind situations.

Figure 19 shows wind roses for the selected points in the Middle and South Caspian (see point location in Figure 1) for the whole period. These roses show that the wind regime is very different in the South and Middle Caspian Sea. It can be seen that the MC is dominated by northwest and south-southwest winds, while the South Caspian is characterized by south winds. It can also be seen that in the Middle Caspian, the winds with higher velocity are more repeatable. In the South Caspian, the probability of more than 6% is distributed around 7–8 directions, while in the Middle Caspian, it is around 6 directions, so the maximum probability is higher in the Middle Caspian. Higher probability of one wind direction suggests a longer duration of this wind, which determines higher wave

heights and agrees well with the number of storms (the number of storms in the Middle Caspian is higher than in the South Caspian).

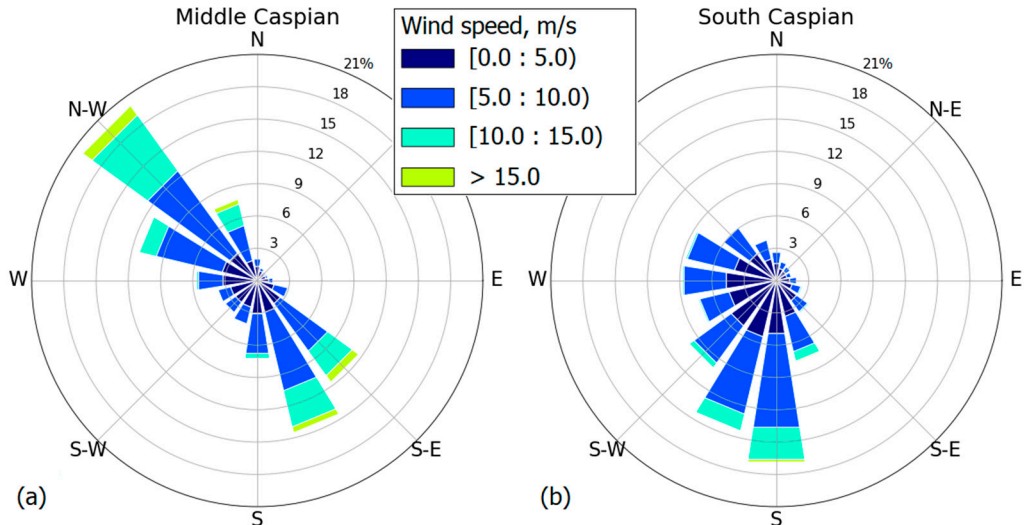

**Figure 19.** Wind roses for the point in the Middle Caspian Sea (**a**) and for the point in the South Caspian Sea (**b**) for the period of 1979–2020.

When analyzing the six-year wind roses (Figures 20 and 21), climate changes in the probability of winds from the leading directions were observed. At the midpoint, there were fewer southeasterly winds from 1991 to 1996, while northwesterly winds increased for the same period.

In the South Caspian point, there was a noticeable increase in the probability of winds from south-southwest and southwest. In 2003–2008, the probability of winds from south-southwest exceeded those from southwest, and since 1991, the number of southwest winds exceeded south-southeast winds.

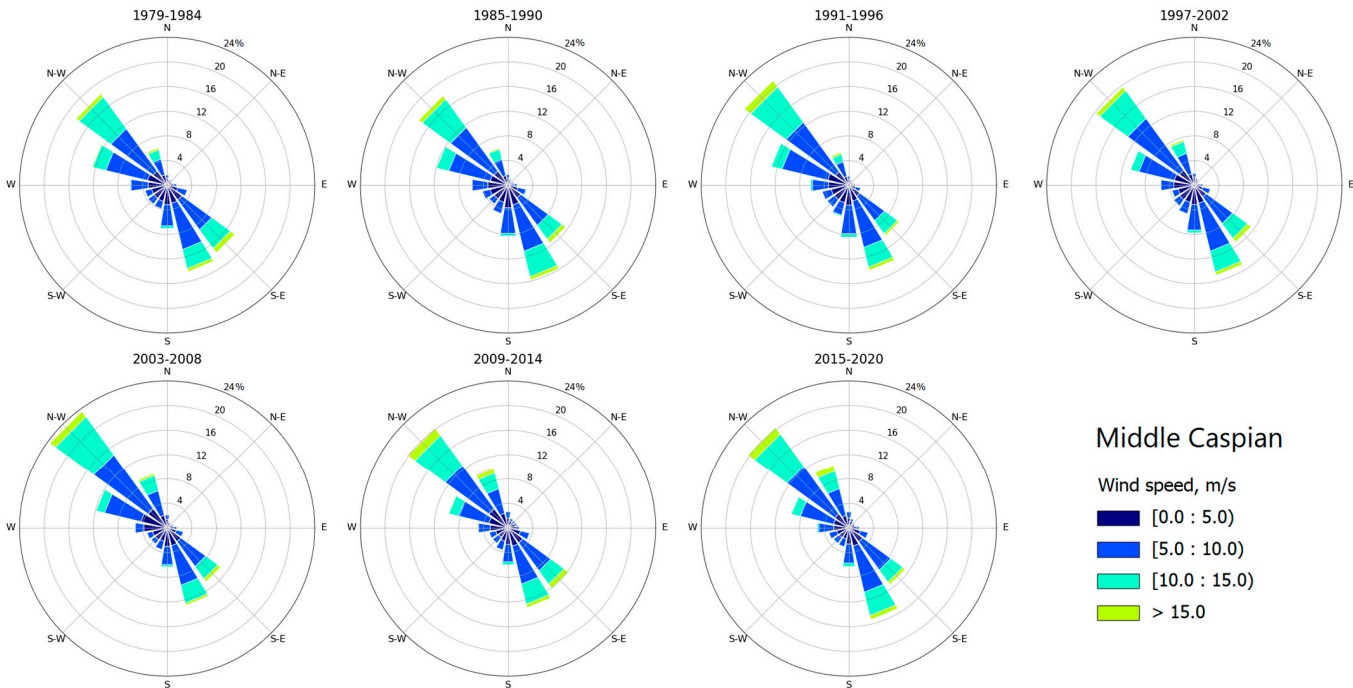

**Figure 20.** Wind rose for the point in the Middle Caspian Sea for six-year periods.

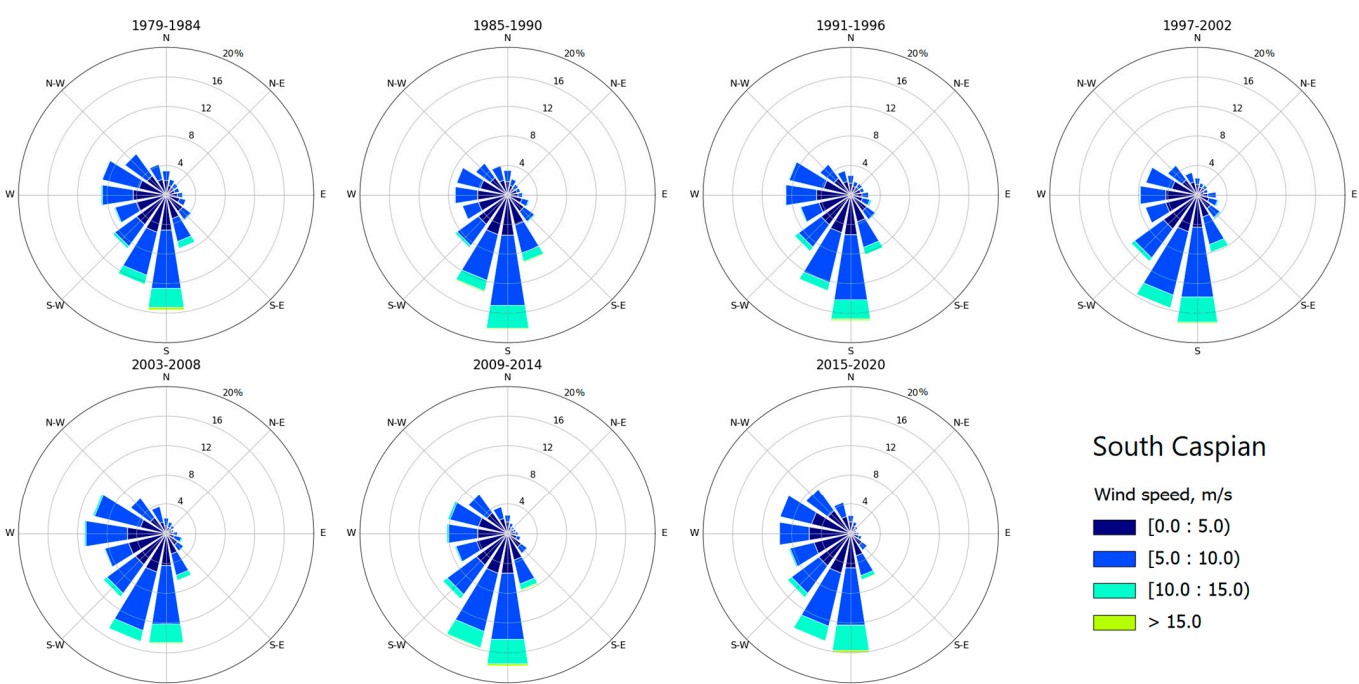

**Figure 21.** Wind roses for the point in the South Caspian Sea for six-year periods.

Figure 22 shows the 95th and 5th percentiles (as components U and V can be positive and negative) in the Middle Caspian Sea. There is an increase in the 95th percentile component V. It is worth noting that the 5th percentiles are not subject to strong changes for both components. The positive trend for the 95th percentile of wind speed component V is significant (Figure 22).

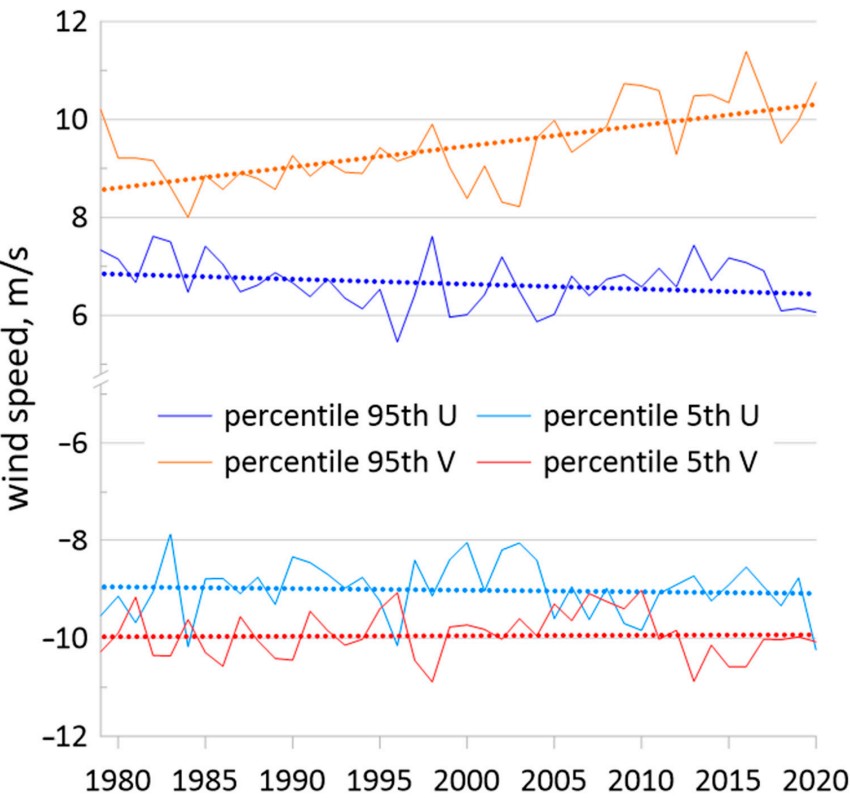

**Figure 22.** The 95th and 5th percentiles for the wind speed components at the point of the Middle Caspian Sea. The dotted lines indicate the linear trends.

A very different picture is presented at the South Caspian point (Figure 23). Both the 95th and the 5th percentiles decrease for the U and V components. However, the 5th percentile decrease for the U and V components mean that wind speed increases. The negative trends were significant for the 95th percentile component U, and the trend was −0.3 m/s in 10 years; for the 5th percentile component U, the trend was −0.1 m/s in 10 years. The trends for the V component were not significant.

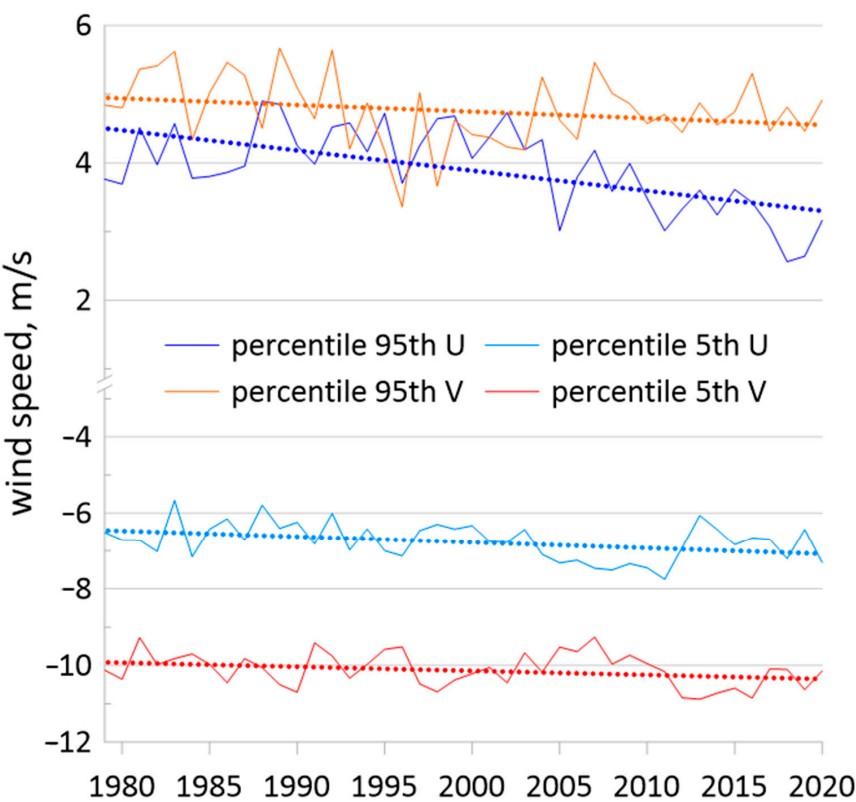

**Figure 23.** The 95th and 5th percentiles for the wind speed components at the point of the South Caspian Sea. The dotted lines indicate the linear trends.

It can be seen from the roses that in the Middle Caspian, the winds began to blow more often from the northwestern sector. This can also be confirmed in publications [50,51]. In the same way, the 95th percentile of V component of wind increased, which indicates that besides the frequency of these directions, their maximum velocities also increased. In the South Caspian Sea, in contrast to the Middle Caspian, we observed a decrease in the 95th percentile, but the 5th percentile increased (in absolute values).

Next, the annual mean percentiles (80th to 95th) of wind speed module ($W^2 = U^2 + V^2$) were calculated to find the relationship between changes in the number of storms and changes in wind speed. As a result, a correlation analysis was carried out for the annual mean, 95th and 5th percentiles of the wind speed components, and the number of storms in the Middle and the South Caspian Seas (Table 4).

As shown in Table 4, the highest correlation coefficient was found between the number of storms and the 95th percentile wind speeds. The correlation of 0.6 between the number of storms and the mean annual U component was observed only in the South Caspian Sea. There was also a high correlation between the number of storms and the 5th percentile V component of wind speed in the South Caspian Sea. At the same time, a high positive correlation was observed for the 95th percentile of V-component wind speed and negative correlation with the 5th percentile of V-component wind speed in the Middle Caspian.

As a result of this section, we found high correlations (0.7–0.8) between the number of storms and wind events > 12–14 m/s and with duration of 6–9 h in the Middle and South Caspian. There is also a correlation between the number of storms and the 95th percentile

wind speed module and U and V components. This relationship allows us to try to explain the different direction of trends in the MC and SC.

**Table 4.** Correlation analysis for the wind speed components and number of storms in the Middle and South Caspian. Red shading indicates $R^2 > 0.5$; blue shading indicates $R^2 < -0.5$.

| | Wind Speed ($W^2 = U^2 + V^2$) | | | | Mean Annual Values | | U Component | | V Component | |
|---|---|---|---|---|---|---|---|---|---|---|
| | 80th Per. | 85th Per. | 90th Per. | 95th Per. | U | V | 5th Per. | 95th Per. | 5th Per. | 95th Per. |
| Storms in the Middle Caspian Sea | 0.55 >3 m | 0.63 >3 m | 0.65 >3 m | 0.73 >3 m | 0.29 >2 m | 0.18 >4 m | −0.12 >5 m | 0.43 >5 m | −0.63 >3 m | 0.66 >4 m |
| Storms in the South Caspian Sea | 0.43 >2 m | 0.50 >2 m | 0.54 (>2 m) | 0.66 >3 m | 0.60 >3 m | −0.45 >3 m | 0.48 >3 m | 0.36 >2 m | −0.58 (>3 m) | 0.09 >2 m |

The positive trends in the number of storms with SWH > 2–3 m in the MC are almost completely explained by positive trends in the 95th percentile of wind speed module and V component (Figure 24a,b). For the South Caspian Sea, we see that the 10–12 m/s wind events and the 95th percentile wind speed have neutral or negative and insignificant trends (Figure 24c,d). In the SC, the strongest wind has a south direction, which leads to a small fetch limitation, and swell waves cannot be generated. Most likely, this is why a slight increase in the 5th percentile wind speed V component in the SC does not lead to an increase in the number of storms.

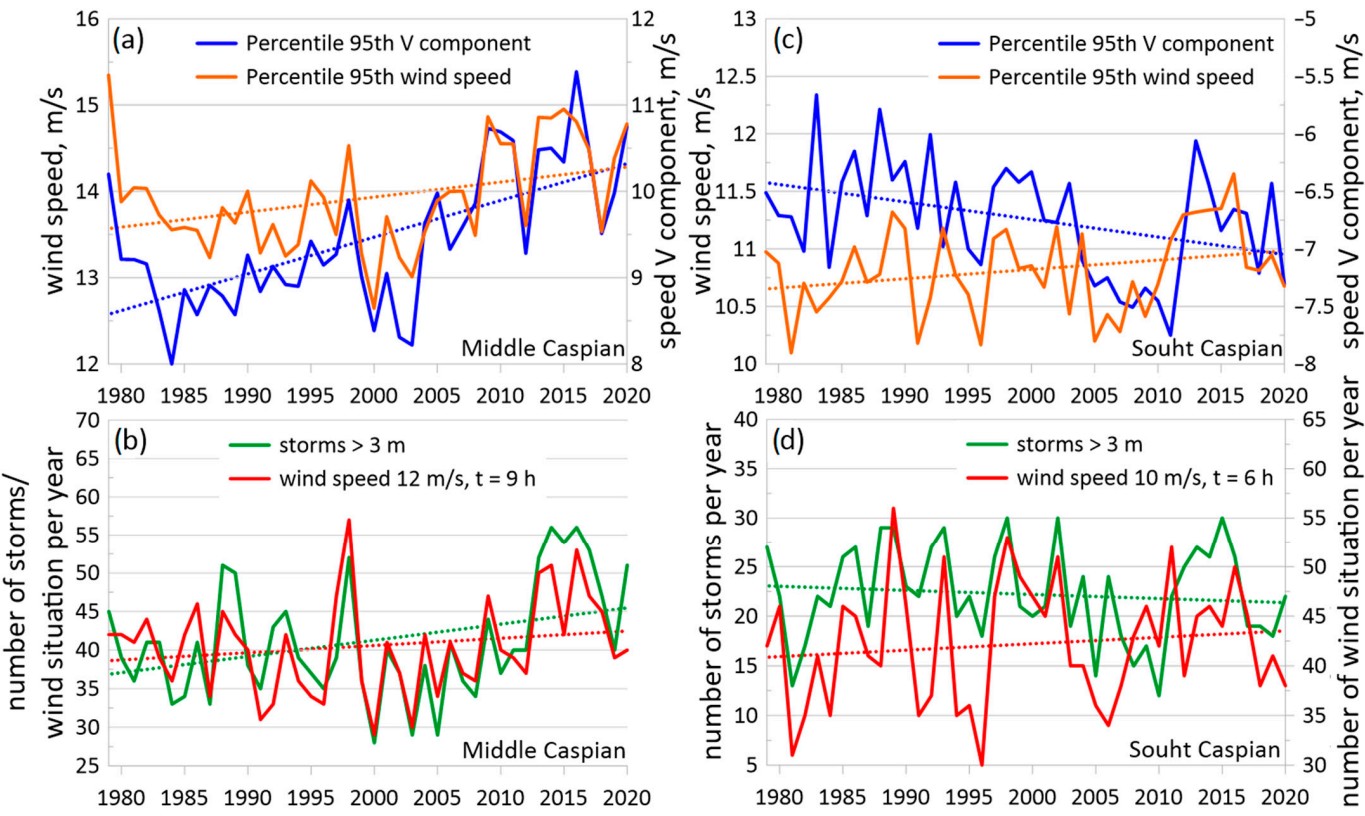

**Figure 24.** Percentiles of wind speed, number of storms, and wind events in the Middle Caspian (**a**,**b**) and in the South Caspian (**c**,**d**). The dotted lines indicate the linear trends.

## 4. Conclusions

The maximum SWH = 8.17 m and the maximum long-term mean SWH ~1.5 m were observed in the Middle Caspian in winter. In the South Caspian in spring and autumn, the local maximum SWH was observed in spring, < 7 m, and in autumn, < 7.5 m. In summer, the maximum SWH did not exceed 6 m and the long-term mean did not exceed 1 m. In this work, we add to our previous results the analysis of the 95th percentile of SWH and seasonal approaches.

Using the POT method, the number of storm events per year with SWH 2–5 m was analyzed in the multiyear and seasonal approaches for the whole Caspian Sea. The mean multiyear number of storms with SWH > 2 m was 90. Significant positive trends were found for the number of storms with SWH > 3–4 m.

The assessment of the total duration of storms in the Caspian Sea is also innovative. A significant positive trend was found in total storm duration: every 10 years, the number of stormy days with SWH > 2 m increases by 4 days. The total storm duration with SWH > 2 m decreases in summer. Significant trends in storm duration were observed in spring and winter.

Storm trends in various parts of the Caspian Sea have not been assessed in research works previously. Positive significant trends were found in the North Caspian (for SWH > 3 m and total storm duration SWH > 2–3 m) and in the Middle Caspian (for SWH > 3–4 m and total storm duration SWH > 3 m). No significant trends were found in the South Caspian, but trends were negative.

For a more detailed analysis, points in the Middle and South Caspian Sea were selected. The selected points reflect well the changes in storm activity in the areas—the Middle Caspian is characterized by positive significant trends, while the South Caspian is characterized by negatively directed trends.

We proposed an original approach to assess the relationship between the climatic changes in wind speed variability and storm wave recurrence. We found high correlations (0.7–0.8) between the number of storms and wind events > 12–14 m/s and the 95th percentile wind speed in the Middle and South Caspian, which allows us to explain the trends in the number of storms.

The positive trends in the number of storms with SWH > 2–3 m in the MC can almost completely be explained by positive trends in the 95th percentile of wind speed. For the South Caspian Sea, we see that the 10–12 m/s wind events and the 95th percentile wind speed have neutral or negative and insignificant trends, which can explain negative trends in the number of storms. In addition, some fetch limitation might prevent an increase in the number of storms when the increase in the 95th percentile of south wind speed is observed.

The results obtained in this article showed that even for such a small body of water as the Caspian Sea, there can be multidirectional trends in the recurrence of storm waves in different parts of the sea. An original approach was also proposed to assess the relationship between climatic wind speed variability and the recurrence of storm waves. The obtained results can be used in various climate studies of the Caspian Sea, and the proposed approaches can be applied to other seas.

**Author Contributions:** The concept of the study was developed by S.M. S.M. performed numerical simulations, analysis, visualization, and manuscript writing. E.K. performed the probability analysis of the storm waves, analysis, visualization, and manuscript writing. E.K. prepared the paper with contributions from S.M. All authors have read and agreed to the published version of the manuscript.

**Funding:** The works of S.A. Myslenkov were supported by the Interdisciplinary Scientific and Educational School of Moscow State University "The Future of the Planet and Global Environmental Changes". Data analysis was funded by the Ministry of Science and Higher Education of Russia, theme FMWE-2021-0002 E.E. Kruglova.

**Data Availability Statement:** The wind wave general statistics which presented in this article provided in open access (https://carto.geogr.msu.ru/wavenergy/; http://93.180.9.222/wavenergy, accessed on 25 May 2023). The raw data of the wave calculations are not available in open access.

**Conflicts of Interest:** The authors declare that the research was conducted in the absence of any commercial or financial relationship that could be considered a potential conflict of interest.

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
