# Peer review of "Influence of Long-Term Wind Variability on the Storm Activity in the Caspian Sea"

_water, doi:10.3390/w15112125_

Round 1

Reviewer 1 Report

The manuscript is devoted to assessing the influence of wind variability on storm activity in the Caspian Sea. A lot of work has been done to process storm wave data for the entire Caspian Sea from 1979 to 2020. It is a good manuscript, but has a few minor comments.

1. Poor quality of figures.

2. In most figures, the signatures on the axes are very small.

3. The article does not describe how the heights of storm waves are related to the depth and relief of the bottom of the observation zones. Have such assessments been made?

4. The article describes that the correlation coefficient of comparison of model and field data is 0.92. But judging by the graph shown in Figure 2a, it should be much lower.

5. In conclusion, it would be good to add a few words about the scope of the results obtained.

The manuscript is devoted to assessing the influence of wind variability on storm activity in the Caspian Sea. A lot of work has been done to process storm wave data for the entire Caspian Sea from 1979 to 2020. It is a good manuscript, but has a few minor comments.

1. Poor quality of figures.

2. In most figures, the signatures on the axes are very small.

3. The article does not describe how the heights of storm waves are related to the depth and relief of the bottom of the observation zones. Have such assessments been made?

4. The article describes that the correlation coefficient of comparison of model and field data is 0.92. But judging by the graph shown in Figure 2a, it should be much lower.

5. In conclusion, it would be good to add a few words about the scope of the results obtained.

Author Response

We express a great gratitude to you for your review, careful study of the materials of the article and constructive comments. Our point-to-point response to your comments presented below.

Author's Reply to the Review Report (Reviewer 1)

Comments and Suggestions for Authors

The manuscript is devoted to assessing the influence of wind variability on storm activity in the Caspian Sea. A lot of work has been done to process storm wave data for the entire Caspian Sea from 1979 to 2020. It is a good manuscript, but has a few minor comments.

  1. Poor quality of figures.

The quality of all figures been improved

  1. In most figures, the signatures on the axes are very small.

The font of the axes and legends have been enlarged for most figures.

  1. The article does not describe how the heights of storm waves are related to the depth and relief of the bottom of the observation zones. Have such assessments been made?

In this paper, we did not evaluate directly the influence of the bottom depth and relief on the wave heights. Only in the North part of the Caspian Sea the bottom relief can influence to the wind waves. Middle and South parts of the Caspian Sea is a “deep water” in term of wind modelling. In this article we focus to the storm waves trend analysis in deep Middle and South parts of the Caspian Sea. We add a small remark to the introduction about bottom relief influence. (lines 38-44)

  1. The article describes that the correlation coefficient of comparison of model and field data is 0.92. But judging by the graph shown in Figure 2a, it should be much lower.

Most likely you are right, but unfortunately, we had only the image of wave measurements which published in [Ambrosimov, 2008] and we can not calculate statistics without digital data. The correlation coefficient 0.92 was calculated only for the comparison of model and altimeter data, in Figure 2 b. We add the text with explanations of this situation in lines 106-110

  1. In conclusion, it would be good to add a few words about the scope of the results obtained.

In conclusion, a paragraph was added (lines 523-528)

Reviewer 2 Report

No comments.

Author Response

We express a great gratitude to you for your review

Reviewer 3 Report

The main objective of this study is to assess the impact of sea breeze changes on storm activity in the Caspian Sea over the past 42 years. The WAVEWATCH III and peak-threshold analysis of storm quantity results are presented. The statistical data of wave height are analyzed year by year and season by season, and the changing trend of storms in various regions of the Caspian Sea is analyzed. It concludes that the number and duration of storms is rising in the North and Middle Caspian Seas, while the opposite is true in the South Caspian Sea. The research of this paper is of great significance for storm prediction and disaster prevention in the Caspian Sea.

Although this paper has practical significance for the prediction of the Caspian Sea, but this paper describe the original content fewly. The numerical model method for Caspian storm in this paper is only an application of other people's existing research results, which is insufficient to reflect the part of innovation. So it is not recommended to publish on water. It is suggested to re-submit the article after upgrading or try to submit it to other journals.

Author Response

We express a great gratitude to you for your review, careful study of the materials of the article and constructive comments. Our point-to-point response to your comments presented below.

Author's Reply to the Review Report (Reviewer 3)

Comments and Suggestions for Authors

The main objective of this study is to assess the impact of sea breeze changes on storm activity in the Caspian Sea over the past 42 years. The WAVEWATCH III and peak-threshold analysis of storm quantity results are presented. The statistical data of wave height are analyzed year by year and season by season, and the changing trend of storms in various regions of the Caspian Sea is analyzed. It concludes that the number and duration of storms is rising in the North and Middle Caspian Seas, while the opposite is true in the South Caspian Sea. The research of this paper is of great significance for storm prediction and disaster prevention in the Caspian Sea.

Although this paper has practical significance for the prediction of the Caspian Sea, but this paper describe the original content fewly. The numerical model method for Caspian storm in this paper is only an application of other people's existing research results, which is insufficient to reflect the part of innovation. So it is not recommended to publish on water. It is suggested to re-submit the article after upgrading or try to submit it to other journals.

You are right - we used the wind wave database obtained earlier and we published the results about the wave climate of the Caspian Sea in our earlier articles (Myslenkov S. et al., 2018, Pavlova A., Myslenkov S., et al., 2022). Many scientists use open databases of wave parameters, for example, from ERA 5 for climate researches. In our previous article (Pavlova A., Myslenkov S., et al., 2022) only the number of storms for the whole Caspian Sea was presented without any trend analysis.

But in this article, we present only new and original results about the storm activity in the Caspian Sea which have not been published before:

  • Detailed analysis of storm activity trends for different parts of the Caspian Sea
  • Assessment of seasonal variability of storm waves recurrence (for the entire Caspian Sea and for various parts)
  • New results about the total duration of storms are presented
  • An important conclusion was obtained about the winter-spring period makes the greatest contribution to the variability of storm waves recurrence
  • Very important results which devoted to the multidirectional trends in different parts of the Caspian Sea was obtained
  • We proposed the original an approach to assess the relationship between climatic changes in wind speed variability and the storm waves recurrence
  • The reasons for the change in the number of storms in different parts of the Caspian Sea and the divergence of their trends have not been analyzed before
  • In our work the detail analysis of wind direction changes described. Wind direction changes does not make a significant contribution to the storm waves recurrence
  • The analysis of wind situations showed the presence of significant trends, as well as the trends of wind percentiles and wind components V and U. This can help us to explain the founded multidirectional trends in storm activity.

We add to manuscript new additions to describe clearer the earlier results and new achievements in lines:

72-80; 188-190; 216-218; 257-258; 491-493; 495; 503-504; 512-513

Round 2

Reviewer 3 Report

The main objective of this study is to assess the impact of sea breeze changes on storm activity in the Caspian Sea over the past 42 years. The WAVEWATCH III and Peak Over Threshold method analysis of storm quantity results are presented. The statistical data of wave height are analyzed year by year and season by season, and the changing trend of storms in various regions of the Caspian Sea is analyzed. It concludes that the number and duration of storms is rising in the North and Middle Caspian Seas, while the opposite is true in the South Caspian Sea. The research of this paper is of great significance for storm prediction and disaster prevention in the Caspian Sea.

The addition and modification of this paper did not change the research content of the article itself, and the originality of the research content still did not increase. The subject remains the use of others' work. The part of innovation is not enough. So it is not recommended to publish on water. It is suggested to re-submit the article after upgrading or try to submit it to other journals.

Nothing.

Author Response

Thank you for your comments and recommendations.